# SPATIALLY TRANSFORMED ADVERSARIAL EXAMPLES

**Chaowei Xiao**[1] [*], **Jun-Yan Zhu**[2] [*], **Bo Li**[3], **Warren He**[3], **Mingyan Liu**[1], **Dawn Song**[3]
[1]University of Michigan, Ann Arbor, USA
[2]Massachusetts Institute of Technology, MA, USA
[3]University of California, Berkeley, USA

## ABSTRACT

Recent studies show that widely used deep neural networks (DNNs) are vulnerable to carefully crafted adversarial examples. Many advanced algorithms have been proposed to generate adversarial examples by leveraging the $\mathcal{L}_p$ distance for penalizing perturbations. Researchers have explored different defense methods to defend against such adversarial attacks. While the effectiveness of $\mathcal{L}_p$ distance as a metric of perceptual quality remains an active research area, in this paper we will instead focus on a different type of perturbation, namely spatial transformation, as opposed to manipulating the pixel values directly as in prior works. Perturbations generated through spatial transformation could result in large $\mathcal{L}_p$ distance measures, but our extensive experiments show that such spatially transformed adversarial examples are perceptually realistic and more difficult to defend against with existing defense systems. This potentially provides a new direction in adversarial example generation and the design of corresponding defenses. We visualize the spatial transformation based perturbation for different examples and show that our technique can produce realistic adversarial examples with smooth image deformation. Finally, we visualize the attention of deep networks with different types of adversarial examples to better understand how these examples are interpreted.

## 1   INTRODUCTION

Deep neural networks (DNNs) have demonstrated their outstanding performance in different domains, ranging from image processing (Krizhevsky et al., 2012; He et al., 2016), text analysis (Collobert & Weston, 2008) to speech recognition (Hinton et al., 2012). Though deep networks have exhibited high performance for these tasks, recently they have been shown to be particularly vulnerable to adversarial perturbations added to the input images (Szegedy et al., 2013; Goodfellow et al., 2015). These perturbed instances are called *adversarial examples*, which can lead to undesirable consequences in many practical applications based on DNNs. For example, adversarial examples can be used to subvert malware detection, fraud detection, or even potentially mislead autonomous navigation systems (Papernot et al., 2016b; Evtimov et al., 2017; Grosse et al., 2016) and therefore pose security risks when applied to security-related applications. A comprehensive study about adversarial examples is required to motivate effective defenses. Different methods have been proposed to generate adversarial examples such as fast gradient sign methods (FGSM) (Goodfellow et al., 2015), which can produce adversarial instances rapidly, and optimization-based methods (C&W) (Carlini & Wagner, 2017a), which search for adversarial examples with smaller magnitude of perturbation.

One important criterion for adversarial examples is that the perturbed images should "look like" the original instances. The traditional attack strategies adopt $L_2$ (or other $\mathcal{L}_p$) norm distance as a perceptual similarity metric to evaluate the distortion (Gu & Rigazio, 2014). However, this is not an ideal metric (Johnson et al., 2016; Isola et al., 2017), as $L_2$ similarity is sensitive to lighting and viewpoint change of a pictured object. For instance, an image can be shifted by one pixel, which will lead to large $L_2$ distance, while the translated image actually appear "the same" to human perception. Motivated by this example, in this paper we aim to look for other types of adversarial examples and propose to create perceptually realistic examples by changing the positions of pixels instead

---

[*]indicates equal contributions

of directly manipulating existing pixel values. This has been shown to better preserve the identity and structure of the original image (Zhou et al., 2016b). Thus, the proposed spatially transformed adversarial example optimization method (stAdv) can keep adversarial examples less distinguishable from real instances (such examples can be found in Figure 3).

Various defense methods have also been proposed to defend against adversarial examples. Adversarial training based methods have so far achieved the most promising results (Goodfellow et al., 2015; Tramèr et al., 2017; Mądry et al., 2017). They have demonstrated the robustness of improved deep networks under certain constraints. However, the spatially transformed adversarial examples are generated through a rather different principle, whereby what is being minimized is the local geometric distortion rather than the $\mathcal{L}_p$ pixel error between the adversarial and original instances. Thus, the previous adversarial training based defense method may appear less effective against this new attack given the fact that these examples generated by stAdv have never been seen before. This opens a new challenge about how to defend against such attacks, as well as other attacks that are not based on direct pixel value manipulation.

We visualize the spatial deformation generated by stAdv; it is seen to be locally smooth and virtually imperceptible to the human eye. In addition, to better understand the properties of deep neural networks on different adversarial examples, we provide visualizations of the attention of the DNN given adversarial examples generated by different attack algorithms. We find that the spatial transformation based attack is more resilient across different defense models, including adversarially trained robust models.

Our contributions are summarized as follows:

- We propose to generate adversarial examples based on spatial transformation instead of direct manipulation of the pixel values, and we show realistic and effective adversarial examples on the MNIST, CIFAR-10, and ImageNet datasets.
- We provide visualizations of optimized transformations and show that such geometric changes are small and locally smooth, leading to high perceptual quality.
- We empirically show that, compared to other attacks, adversarial examples generated by stAdv are more difficult to detect with current defense systems.
- Finally, we visualize the attention maps of deep networks on different adversarial examples and demonstrate that adversarial examples based on stAdv can more consistently mislead the adversarial trained robust deep networks compared to other existing attack methods.

## 2 RELATED WORK

Here we first briefly summarize the existing adversarial attack algorithms as well as the current defense methods. We then discuss the spatial transformation model used in our adversarial attack.

**Adversarial Examples**    Given a benign sample $\mathbf{x}$, an attack instance $\mathbf{x}_{adv}$ is referred to as an adversarial example, if a small magnitude of perturbation $\epsilon$ is added to $\mathbf{x}$ (i.e. $\mathbf{x}_{adv} = \mathbf{x} + \epsilon$) so that $\mathbf{x}_{adv}$ is misclassified by the targeted classifier $g$. Based on the adversarial goal, attacks can be classified into two categories: targeted and untargeted attacks. In a targeted attack, the adversary's objective is to modify an input $\mathbf{x}$ such that the target model $g$ classifies the perturbed input $\mathbf{x}_{adv}$ in a *targeted* class chosen, which differs from its ground truth. In a untargeted attack, the adversary's objective is to cause the perturbed input $\mathbf{x}_{adv}$ to be misclassified in *any class* other than its ground truth. Based on the adversarial capabilities, these attacks can be categorized as white-box and black-box attacks, where an adversary has full knowledge of the classifier and training data in the white-box setting (Szegedy et al., 2014; Goodfellow et al., 2015; Carlini & Wagner, 2017a; Moosavi-Dezfooli et al., 2015; Papernot et al., 2016b; Biggio et al., 2013; Fawzi & Frossard, 2015; Kanbak, 2017; Kurakin et al., 2016); while having zero knowledge about them in the black-box setting (Papernot et al., 2016a; Liu et al., 2017; Moosavi-Dezfooli et al., 2016; Mopuri et al., 2017). In this work, we will focus on the white-box setting to explore what a powerful adversary can do based on the Kerckhoffs's principle (Shannon, 1949) to better motivate defense methods.

**Spatial Transformation**    In computer vision and graphics literature, Two main aspects determine the appearance of a pictured object (Szeliski, 2010): (1) the *lighting and material*, which determine

the brightness of a point as a function of illumination and object material properties, and (2) the *geometry*, which determines where the projection of a point will be located in the scene. Most previous adversarial attacks (Goodfellow et al., 2015) focus on changing the *lighting and material* aspect, while assuming the underlying geometry stays the same during the adversarial perturbation generation process.

Modeling geometric transformation with neural networks was first explored by "capsules," computational units that locally transform their input for modeling 2D and 3D geometric changes (Hinton et al., 2011). Later, Jaderberg et al. (2015) demonstrated that similar computational units, named spatial transformers, can benefit many visual recognition tasks. Zhou et al. (2016a) adopted the spatial transformers for synthesizing novel views of the same object and has shown that a geometric method can produce more realistic results compared to pure pixel-based methods. Inspired by these successes, we also use the spatial transformers to deform the input images, but with a different goal: to generate realistic adversarial examples.

**Defensive Methods**   Following the emergence of adversarial examples, various defense methods have been studied, including adversarial training (Goodfellow et al., 2015), distillation (Papernot et al., 2016c), gradient masking (Gu & Rigazio, 2014) and feature squeezing (Xu et al., 2017). However, these defenses can either be evaded by C&W attacks or only provide marginal improvements (Carlini & Wagner, 2017b; He et al., 2017). Among these defenses, adversarial training has achieved the state-of-the-art performance. Goodfellow et al. (2015) proposed to use the fast gradient sign attack as an adversary to perform adversarial training, which is much faster, followed by ensemble adversarial training (Tramèr et al., 2017) and projected gradient descent (PGD) adversarial training (Mądry et al., 2017). In this work, we explicitly analyze how effective the spatial transformation based adversarial examples are under these adversarial training based defense methods.

## 3   GENERATING ADVERSARIAL EXAMPLES

Here we first introduce several existing attack methods and then present our formulation for producing spatially transformed adversarial examples.

### 3.1   PROBLEM DEFINITION

Given a learned classifier $g : \mathcal{X} \to \mathcal{Y}$ from a feature space $\mathcal{X}$ to a set of classification outputs $\mathcal{Y}$ (e.g., $\mathcal{Y} = \{0, 1\}$ for binary classification), an adversary aims to generate adversarial example $\mathbf{x}_{\text{adv}}$ for an original instance $\mathbf{x} \in \mathcal{X}$ with its ground truth label $y \in \mathcal{Y}$, so that the classifier predicts $g(\mathbf{x}_{\text{adv}}) \neq y$ (untargeted attack) or $g(\mathbf{x}_{\text{adv}}) = t$ (targeted attack) where $t$ is the target class.

### 3.2   BACKGROUND: CURRENT PIXEL-VALUE BASED ATTACK METHODS

All of the current methods for generating adversarial examples are built on directly modifying the pixel values of the original image.

The *fast gradient sign method* (FGSM) (Goodfellow et al., 2015) uses a first-order approximation of the loss function to construct adversarial samples for the adversary's target classifier $g$. The algorithm achieves untargeted attack by performing a single gradient ascent step: $\mathbf{x}_{\text{adv}} = \mathbf{x} + \epsilon \cdot \text{sign}(\nabla_{\mathbf{x}} \ell_g(\mathbf{x}, y))$, where $\ell_g(\mathbf{x}, y)$ is the loss function (e.g. cross-entropy loss) used to train the original model $g$, $y$ denotes the ground truth label, and the hyper-parameter $\epsilon$ controls the magnitude of the perturbation. A targeted version of it can be done similarly.

*Optimization-based attack* (C&W) produces an adversarial perturbation for a targeted attack based on certain constraints (Carlini & Wagner, 2017a; Liu et al., 2017) as formulated below:

$$\min ||\boldsymbol{\delta}||_p^2 \quad \text{s.t.} \quad g(\mathbf{x} + \boldsymbol{\delta}) = t \quad \text{and} \quad \mathbf{x} + \boldsymbol{\delta} \in X,$$

where the $\mathcal{L}_p$ norm penalty ensures that the added perturbation $\epsilon$ is small. The same optimization procedure can achieve untargeted attacks with a modified constraint $g(\mathbf{x} + \boldsymbol{\delta}) \neq y$.

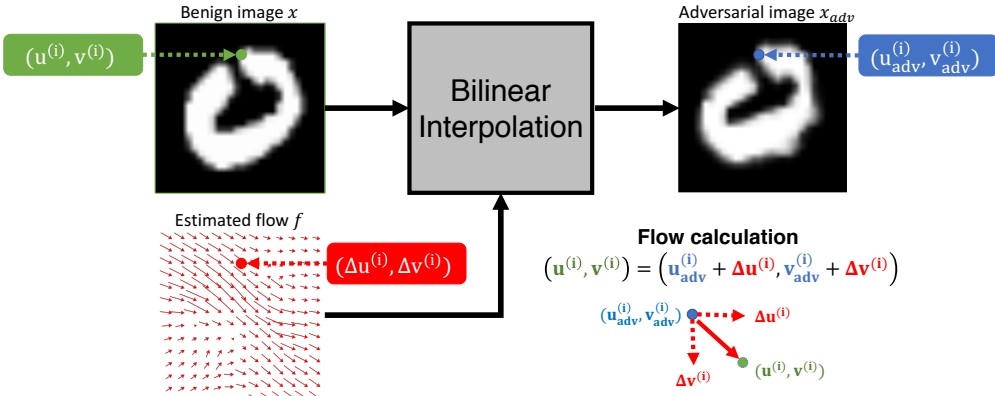

Figure 1: Generating adversarial examples with spatial transformation: the blue point denotes the coordinate of a pixel in an output adversarial image and the green point is its corresponding pixel in an input image. The flow field in red represents the displacement from the pixels in the adversarial image to the pixels in the input image.

## 3.3 OUR APPROACH: SPATIALLY TRANSFORMED ADVERSARIAL EXAMPLES

All the existing approaches directly modify pixel values, which may sometimes produce noticeable artifacts. Instead, we aim to smoothly change the geometry of the scene while keeping the original appearance, producing more perceptually realistic adversarial examples. In this section, we first introduce our spatial transformation model and then describe our objective function for generating spatially transformed adversarial examples.

**Spatial transformation** We use $\mathbf{x}_{\text{adv}}^{(i)}$ to denote the pixel value of the $i$-th pixel and 2D coordinate $(u_{\text{adv}}^{(i)}, v_{\text{adv}}^{(i)})$ to denote its location in the adversarial image $\mathbf{x}_{\text{adv}}$. We assume that $\mathbf{x}_{\text{adv}}^{(i)}$ is transformed from the pixel $\mathbf{x}^{(i)}$ from the original image. We use the per-pixel flow (displacement) field $f$ to synthesize the adversarial image $\mathbf{x}_{\text{adv}}$ using pixels from the input $\mathbf{x}$. For the $i$-th pixel within $\mathbf{x}_{\text{adv}}$ at the pixel location $(u_{\text{adv}}^{(i)}, v_{\text{adv}}^{(i)})$, we optimize the amount of displacement in each image dimension, with the pair denoted by the *flow vector* $f_i := (\Delta u^{(i)}, \Delta v^{(i)})$. Note that the flow vector $f_i$ goes from a pixel $\mathbf{x}_{\text{adv}}^{(i)}$ in the adversarial image to its corresponding pixel $\mathbf{x}^{(i)}$ in the input image. Thus, the location of its corresponding pixel $\mathbf{x}^{(i)}$ can be derived as $(u^{(i)}, v^{(i)}) = (u_{\text{adv}}^{(i)} + \Delta u^{(i)}, v_{\text{adv}}^{(i)} + \Delta v^{(i)})$. As the $(u^{(i)}, v^{(i)})$ can be fractional numbers and does not necessarily lie on the integer image grid, we use the differentiable bilinear interpolation (Jaderberg et al., 2015) to transform the input image with the flow field. We calculate $\mathbf{x}_{\text{adv}}^{(i)}$ as:

$$\mathbf{x}_{\text{adv}}^{(i)} = \sum_{q \in \mathcal{N}(u^{(i)}, v^{(i)})} \mathbf{x}^{(q)}(1 - |u^{(i)} - u^{(q)}|)(1 - |v^{(i)} - v^{(q)}|), \tag{1}$$

where $\mathcal{N}(u^{(i)}, v^{(i)})$ are the indices of the 4-pixel neighbors at the location $(u^{(i)}, v^{(i)})$ (top-left, top-right, bottom-left, bottom-right). We can obtain the adversarial image $\mathbf{x}_{\text{adv}}$ by calculating Equation 1 for every pixel $\mathbf{x}_{\text{adv}}^{(i)}$. Note that $\mathbf{x}_{\text{adv}}$ is differentiable with respect to the flow field $f$ (Jaderberg et al., 2015; Zhou et al., 2016b). The estimated flow field essentially captures the amount of spatial transformation required to fool the classifier.

**Objective function** Most of the previous methods constrain the added perturbation to be small regarding a $\mathcal{L}_p$ metric. Here instead of imposing the $\mathcal{L}_p$ norm on pixel space, we introduce a new regularization loss $\mathcal{L}_{flow}$ on the local distortion $f$, producing higher perceptual quality for adversarial examples. Therefore, the goal of the attack is to generate adversarial examples which can mislead the classifier as well as minimizing the local distortion introduced by the flow field $f$.

Formally, given a benign instance $\mathbf{x}$, we obtain the flow field $f$ by minimize the following objective:

$$f^* = \operatorname*{argmin}_{f} \quad \mathcal{L}_{adv}(x, f) + \tau \mathcal{L}_{flow}(f), \tag{2}$$

where $\mathcal{L}_{adv}$ encourages the generated adversarial examples to be misclassified by the target classifier. $L_{flow}$ ensures that the spatial transformation distance is minimized to preserve high perceptual quality, and $\tau$ balances these two losses.

The goal of $\mathcal{L}_{adv}$ is to guarantee the targeted attack $g(\mathbf{x}_{adv}) = t$ where $t$ is the targeted class, different from the ground truth label $y$. Recall that we transform the input image $\mathbf{x}$ to $\mathbf{x}_{adv}$ with the flow field $f$ (Equation 1). In practice, directly enforcing $g(\mathbf{x}_{adv}) = t$ during optimization is highly non-linear, we adopt the objective function suggested in Carlini & Wagner (2017a).

$$\mathcal{L}_{adv}(x, f) = \max(\max_{i \neq t} g(\mathbf{x}_{adv})_i - g(\mathbf{x}_{adv})_t, \kappa), \tag{3}$$

where $g(x)$ represents the logit output of model $g$, $g(x)_i$ denotes the $i$-th element of the logit vector, and $\kappa$ is used to control the attack confidence level.

To compute $\mathcal{L}_{flow}$, we calculate the sum of spatial movement distance for any two adjacent pixels. Given an arbitrary pixel $p$ and its neighbors $q \in \mathcal{N}(p)$, we enforce the locally smooth spatial transformation perturbation $\mathcal{L}_{flow}$ based on the total variation (Rudin et al., 1992):

$$\mathcal{L}_{flow}(f) = \sum_{p}^{all \; pixels} \sum_{q \in \mathcal{N}(p)} \sqrt{||\Delta u^{(p)} - \Delta u^{(q)}||_2^2 + ||\Delta v^{(p)} - \Delta v^{(q)}||_2^2}. \tag{4}$$

Intuitively, minimizing the spatial transformation can help ensure the high perceptual quality for stAdv, since adjacent pixels tend to move towards close direction and distance. We solve the above optimization with L-BFGS solver (Liu & Nocedal, 1989).

## 4 EXPERIMENTAL RESULTS

In this section, we first show adversarial examples generated by the proposed spatial transformation method and analyze the properties of these examples from different perspectives. We then visualize the estimated flows for adversarial examples and show that with small and smooth transformation, the generated adversarial examples can already achieve a high attack success rate against deep networks. We also show that stAdv can preserve a high attack success rate against current defense methods, which motivates more sophisticated defense methods in the future. Finally, we analyze the attention regions of DNNs, to better understand the attack properties of stAdv.

**Experiment Setup** We set $\tau$ as 0.05 for all our experiments. We use confidence $\kappa = 0$ for both C&W and stAdv for a fair comparison. We leverage L-BFGS (Liu & Nocedal, 1989) as our solver with backtracking linear search.

### 4.1 ADVERSARIAL EXAMPLES BASED ON SPATIAL TRANSFORMATIONS

We show adversarial examples with high perceptual quality for both MNIST (LeCun & Cortes, 1998) and CIFAR-10 (Krizhevsky et al., 2014) datasets.

**stAdv on MNIST** In our experiments, we generate adversarial examples againsts three target models in the white-box setting on the MNIST dataset. Model A, B, and C are derived from the prior work (Tramèr et al., 2017), which represent different architectures. See Appendix A and Table 4 for more details about their network architectures. Table 1 presents the accuracy of pristine MNIST test data on each model as well as the attack success rate of adversarial examples generated by stAdv on these models. Figure 2 shows the adversarial examples against different models where the original instances appear in the diagonal. Each adversarial example achieves a targeted attack, with the target class shown on the top of the column. It is clear that the generated adversarial examples still appear to be in the same class as the original instance for humans. Another advantage for stAdv compared

with traditional attacks is that examples based on stAdv seldom show noise pattern within the adversarial examples. Instead, stAdv smoothly deforms the digits and since such natural deformation also exists in the dataset digits, humans can barely notice such manipulation.

Table 1: Top: accuracy of different models on pristine data (p); bottom: attack success rates of adversarial examples generated by stAdv on MNIST dataset.

| Model | A | B | C |
|---|---|---|---|
| Accuracy (p) | 98.58% | 98.94% | 99.11% |
| Attack Success Rate | 99.95% | 99.98% | 100.00% |

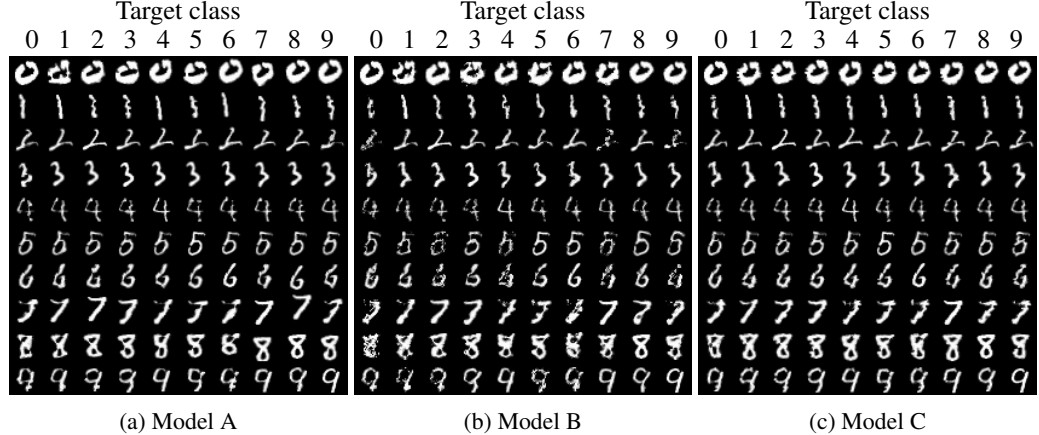

(a) Model A  (b) Model B  (c) Model C

Figure 2: Adversarial examples generated by stAdv against different models on MNIST. The ground truth images are shown in the diagonal and the rest are adversarial examples that are misclassified to the target classes shown on the top.

**stAdv on CIFAR-10**  For CIFAR-10, we use ResNet-32[1] and wide ResNet-34[2] as the target classifiers (Zagoruyko & Komodakis, 2016; He et al., 2016; Mądry et al., 2017). We show the classification accuracy of pristine CIFAR-10 test data (p) and attack success rates of adversarial examples generated by stAdv on different models in Table 2. Figure 3 shows the generated examples on CIFAR-10 against different models. The original images are shown in the diagonal. The other images are targeted adversarial examples, with the index of the target classes shown at the top of the column. Here we use "0-9" to denote the ground truth labels of images lying in the diagonal for each corresponding column. These adversarial examples based on stAdv are randomly selected from the instances that can successfully attack the corresponding classifier. Humans can hardly distinguish these adversarial examples from the original instances.

Table 2: Top: accuracy of different models on pristine data (p); bottom: attack success rates of adversarial examples generated by stAdv on the CIFAR-10 dataset. The numbers in parentheses denote the number of parameters in each target model.

| Model | ResNet32 (0.47M) | Wide ResNet34 (46.16M) |
|---|---|---|
| Accuracy (p) | 93.16% | 95.82% |
| Attack Success Rate | 99.56% | 98.84% |

**Comparison of different adversarial examples**  In Figure 4, we show adversarial examples that are targeted attacked to the same class ("0" for MNIST and "airplane" for CIFAR-10), which is different from their ground truth. We compare adversarial examples generated from different methods

---

[1]https://github.com/tensorflow/models/blob/master/research/ResNet/ResNet_model.py
[2]https://github.com/MadryLab/cifar10_challenge/blob/master/model.py

Target class

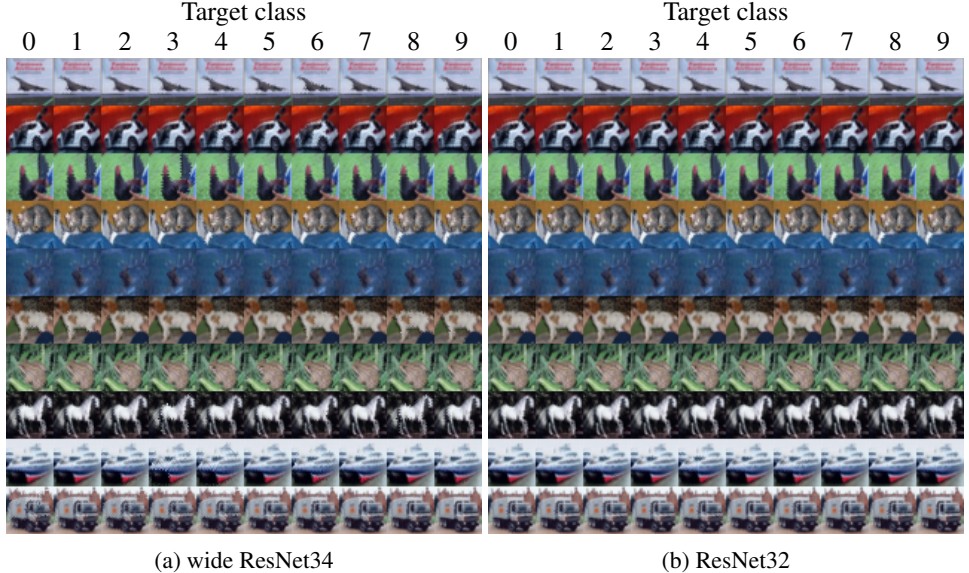

(a) wide ResNet34 (b) ResNet32

Figure 3: Adversarial examples generated by stAdv against different models on CIFAR-10. The ground truth images are shown in the diagonal while the adversarial examples on each column are classified into the same class as the ground truth image within that column.

and show that those based on stAdv look more visually realistic compared with FGSM (Goodfellow et al., 2015) and C&W (Carlini & Wagner, 2017b) methods.

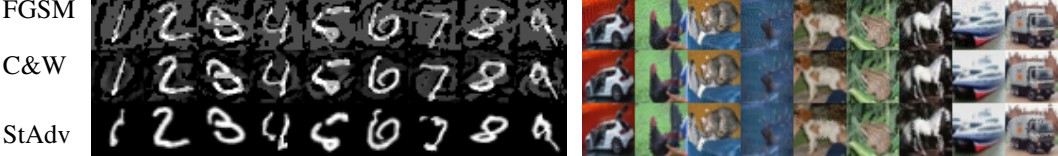

Figure 4: Comparison of adversarial examples generated by FGSM, C&W and stAdv. (Left: MNIST, right: CIFAR-10) The target class for MNIST is "0" and "air plane" for CIFAR-10. We generate adversarial examples by FGSM and C&W with perturbation bounded in terms of $L_\infty$ as 0.3 on MNIST and 8 on CIFAR-10.

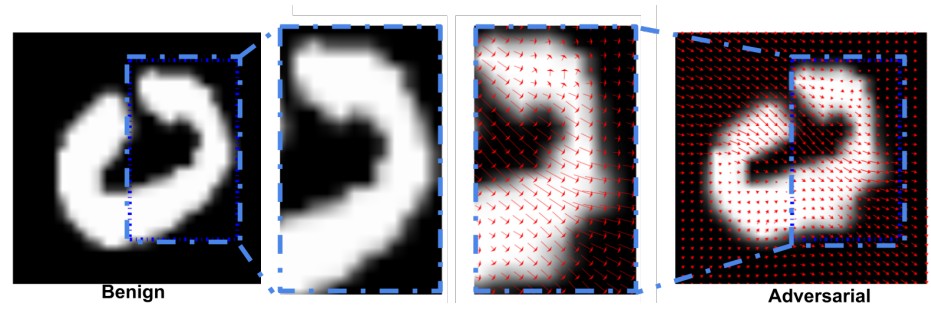

Figure 5: Flow visualization on MNIST. A digit "0" is misclassified as "2".

## 4.2 VISUALIZING SPATIAL TRANSFORMATION

To better understand the spatial transformation applied to the original images, we visualize the optimized transformation flow for different datasets, respectively. Figure 5 visualizes a transformation

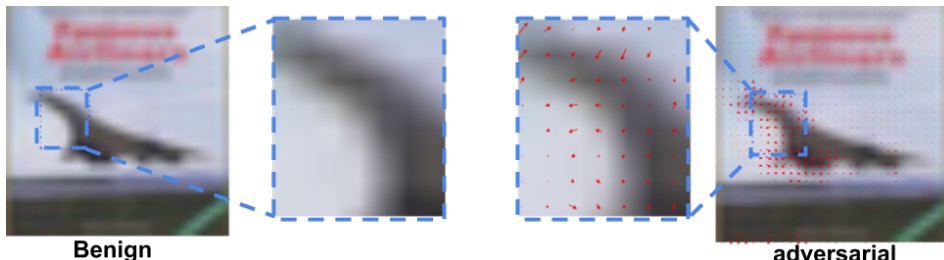

Figure 6: Flow visualization on CIFAR-10. An "airplane" image is misclassified as "bird".

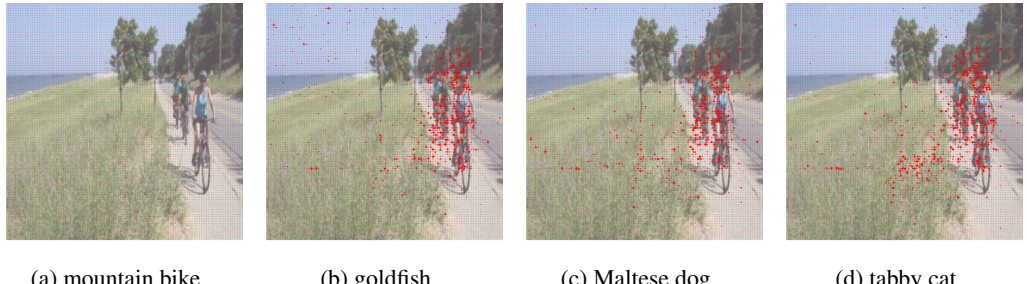

(a) mountain bike        (b) goldfish        (c) Maltese dog        (d) tabby cat

Figure 7: Flow visualization on ImageNet. (a): the original image, (b)-(c): images are misclassified into goldfish, dog and cat, respectively. Note that to display the flows more clearly, we fade out the color of the original image.

on an MNIST instance, where the digit "0" is misclassified as "2." We can see that the adjacent flows move in a similar direction in order to generate smooth results. The flows are more focused on the edge of the digit and sometimes these flows move in different directions along the edge, which implies that the object boundary plays an important role in our stAdv optimization. Figure 6 illustrates a similar visualization on CIFAR-10. It shows that the optimized flows often focus on the area of the main object, such as the airplane. We also observe that the magnitude of flows near the edge are usually larger, which similarly indicates the importance of edges for misleading the classifiers. This observation confirms the observation that when DNNs extract edge information in the earlier layers for visual recognition tasks (Viterbi, 1998). In addition, we visualize the similar flow for the ImageNet dataset (Deng et al., 2009) in Figure 7. The top-1 label of the original image in Figure 7 (a) is "mountain bike". Figure 7 (b)-(d) show targeted adversarial examples generated by stAdv, which have target classes "goldfish," "Maltese dog," and "tabby cat," respectively, and which are predicted as such as the top-1 class. An interesting observation is that, although there are other objects within the image, nearly 90% of the spatial transformation flows tend to focus on the target object bike. Different target class corresponds to different directions for these flows, which still fall into the similar area.

## 4.3 HUMAN PERCEPTUAL STUDY

To quantify the perceptual realism of stAdv's adversarial examples, we perform a user study with human participants on Amazon Mechanical Turk (AMT). We follow the same perceptual study protocol used in prior image synthesis work (Zhang et al., 2016; Isola et al., 2017). We generate 600 images from an ImageNet-compatible dataset, described in Appendix C. In our study, the participants are asked to choose the more *visually realistic* image between an adversarial example generated by stAdv and its original image. During each trial, these two images appear side-by-side for 2 seconds. After the images disappear, our participants are given unlimited time to make their decision. To avoid labeling bias, we allow each user to conduct at most 50 trails. For each pair of an original image and its adversarial example, we collect about 5 annotations from different users.

In total, we collected $2,740$ annotations from 93 AMT users. Examples generated by our method were chosen as the more realistic in $47.01\% \pm 1.96\%$ of the trails (perfectly realistic results would achieve $50\%$). This indicates that our adversarial examples are almost indistinguishable from natural images.

Table 3: Attack success rates of adversarial examples generated by stAdv against models A, B, and C on MNIST, and against ResNet and wide ResNet on CIFAR-10, under standard defenses.

| Model | Def. | FGSM | C&W. | stAdv |
|-------|------|------|------|-------|
| A | Adv. | 4.3% | 4.6% | **32.62%** |
| | Ens. | 1.6% | 4.2% | **48.07%** |
| | PGD | 4.4% | 2.96% | **48.38%** |
| B | Adv. | 6.0% | 4.5% | **50.17%** |
| | Ens. | 2.7% | 3.18% | **46.14%** |
| | PGD | 9.0% | 3.0% | **49.82%** |
| C | Adv. | 3.22% | 0.86% | **30.44%** |
| | Ens. | 1.45% | 0.98% | **28.82%** |
| | PGD | 2.1% | 0.98% | **28.13%** |

| Model | Def. | FGSM | C&W. | stAdv |
|-------|------|------|------|-------|
| ResNet32 | Adv. | 13.10% | 11.9% | **43.36%** |
| | Ens. | 10.00% | 10.3% | **36.89%** |
| | PGD | 22.8% | 21.4% | **49.19%** |
| wide ResNet34 | Adv. | 5.04% | 7.61% | **31.66%** |
| | Ens. | 4.65% | 8.43% | **29.56%** |
| | PGD | 14.9% | 13.90% | **31.6%** |

## 4.4 ATTACK EFFICIENCY UNDER DEFENSE METHODS

Here we generate adversarial examples in the white-box setting and test different defense methods against these samples to evaluate the strength of these attacks under defenses.

We mainly focus on the adversarial training defenses due to their state-of-the-art performance. We apply three defense strategies in our evaluation: the FGSM adversarial training (Adv.) (Goodfellow et al., 2015), ensemble adversarial training (Ens.) (Tramèr et al., 2017), and projectile gradient descent (PGD) adversarial training (Mądry et al., 2017) methods. For adversarial training purposes, we generate adversarial examples based on $L_\infty$ bound (Carlini & Wagner, 2017a) as 0.3 on MNIST and 8 on CIFAR-10. We test adversarial examples generated against model A, B, and C on MNIST as shown in Table 4, and similarly adversarial examples generated against ResNet32 and wide ResNet34 on CIFAR-10.

The results on the MNIST and CIFAR-10 datasets are shown in Table 3. We observe that the three defense strategies can achieve high performance (less than 10% attack success rate) against FGSM and C&W attacks.

These defense methods only achieve low defense performance on stAdv, which improve the attack success rate to more than 30% among all defense strategies. These results indicate that new type of adversarial strategy, such as our spatial transformation-based attack, may open new directions for developing better defense systems. However, for stAdv, we cannot use $\mathcal{L}_p$ norm to bound the distance as translating an image by one pixel may introduce large $\mathcal{L}_p$ penalty. We instead constrain the spatial transformation flow and show that our adversarial examples have high perceptual quality in Figures 2, 3, and 4 as well as Section 4.3.

**Mean blur defense** We also test our adversarial examples against the $3 \times 3$ average pooling restoration mechanism (Li & Li, 2016). Table 5 in Appendix B shows the classification accuracy of recovered images after performing $3 \times 3$ average filter on different models. We find that the simple $3 \times 3$ average pooing restoration mechanism can recover the original class from fast gradient sign examples and improve the classification accuracy up to around 70%. Carlini & Wagner have also shown that such mean blur defense strategy can defend against adversarial examples generated by their attack and improve the model accuracy to around 80% (2017b). From Table 5, we can see that the mean blur defense method can only improve the model accuracy to around 50% on stAdv examples, which means adversarial examples generated by stAdv are more robust compared to other attacks. We also perform a perfect knowledge adaptive attack against the mean blur defense following the same attack strategy suggested in (Carlini & Wagner, 2017b), where we add the $3 \times 3$ average pooling layer into the original network and apply stAdv to attack the new network again. We observe that the success rate of an adaptive attack is nearly 100%, which is consistent with Carlini & Wagner's findings (2017b).

## 4.5 VISUALIZING ATTENTION OF NETWORKS ON ADVERSARIAL EXAMPLES

In addition to the analyzing adversarial examples themselves, in this section, we further characterize these spatially transformed adversarial examples from the perspective of deep neural networks.

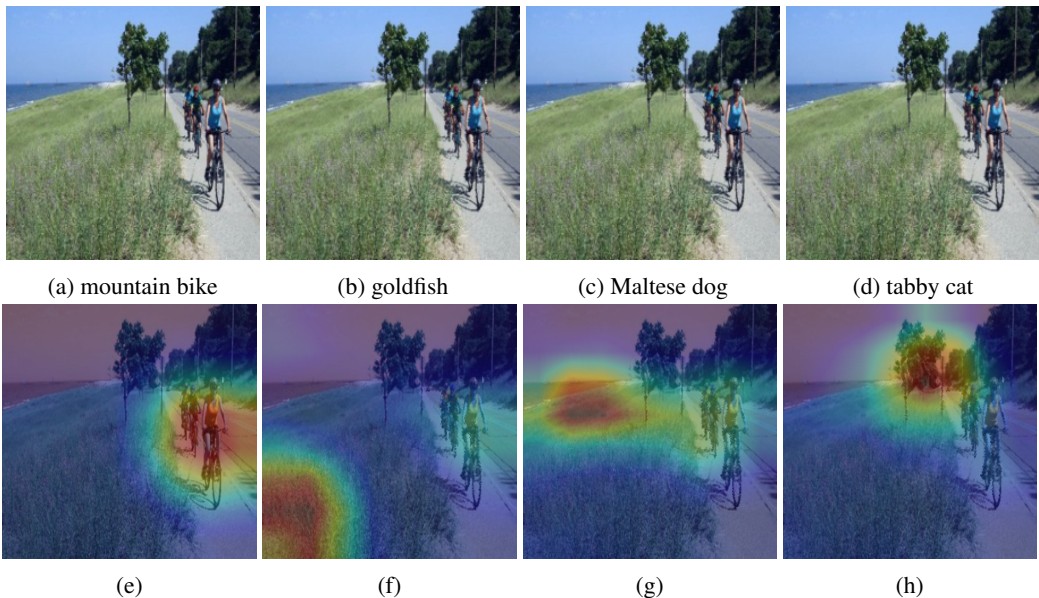

(a) mountain bike      (b) goldfish      (c) Maltese dog      (d) tabby cat

(e)           (f)           (g)           (h)

Figure 8: CAM attention visualizations for ImageNet inception_v3 model. (a) the original image and (b)-(d) stAdv adversarial examples targeting different classes. The second row shows the attention visualizations for the corresponding images displayed above.

Here we apply Class Activation Mapping (CAM) (Zhou et al., 2016a), an implicit attention visualization technique for localizing the discriminative regions implicitly detected by a DNN. We use it to show the attention of the target ImageNet inception_v3 model (Szegedy et al., 2016)) for both original images and generated adversarial examples. Figure 8(a) shows an input bike image and Figure 8(b)–(d) show the targeted adversarial examples based on stAdv targeting three different classes (goldfish, dog, and cat). Figure 8(e) illustrates that the target model draws attention to the bicycle region. Interestingly, attention regions on examples generated by stAdv varies for different target classes as shown in Figure 8(f)–(h). Though humans can barely distinguish between the original image and the ones generated by stAdv, CAM map focus on completely different regions, implying that our attack can mislead the network's attention.

In addition, we also compare and visualize the attention regions of both naturally trained and the adversarial trained inception_v3 model[3] on adversarial images generated by different attack algorithms (Figure 9). The ground truth top-1 label is "cinema," so the attention region for the original image (Figure 9 (a)) includes both tower and building regions. However, when the adversarial examples are targeted attacked into the adversarial label "missile," the attention region focuses on only the tower for all the attack algorithms as shown in Figure 9 (b)-(d) with slight different attention region sizes. More interestingly, we also test these adversarial examples on the public adversarial trained robust inception_v3 model. The result appears in Figure 9 (f)–(h). This time, the attention regions are drawn to the building again for both FGSM and C&W methods, which are close to the attention regions of the original image. The top-1 label for Figure 9 (f) and (g) are again the ground truth "cinema", which means both FGSM and C&W fail to attack the robust model. However, Figure 9 (h) is still misclassified as "missile" under the robust model and the CAM visualization shows that the attention region still focuses on the tower. This example again implies that adversarial examples generated by stAdv are challenging to defend for the current "robust" ImageNet models.

## 5 CONCLUSIONS

Different from the previous works that generate adversarial examples by directly manipulating pixel values, in this work we propose a new type of perturbation based on spatial transformation, which aims to preserve high perceptual quality for adversarial examples. We have shown that adversarial examples generated by stAdv are more difficult for humans to distinguish from original instances.

---

[3]https://github.com/tensorflow/cleverhans/tree/master/examples/nips17_adversarial_competition/

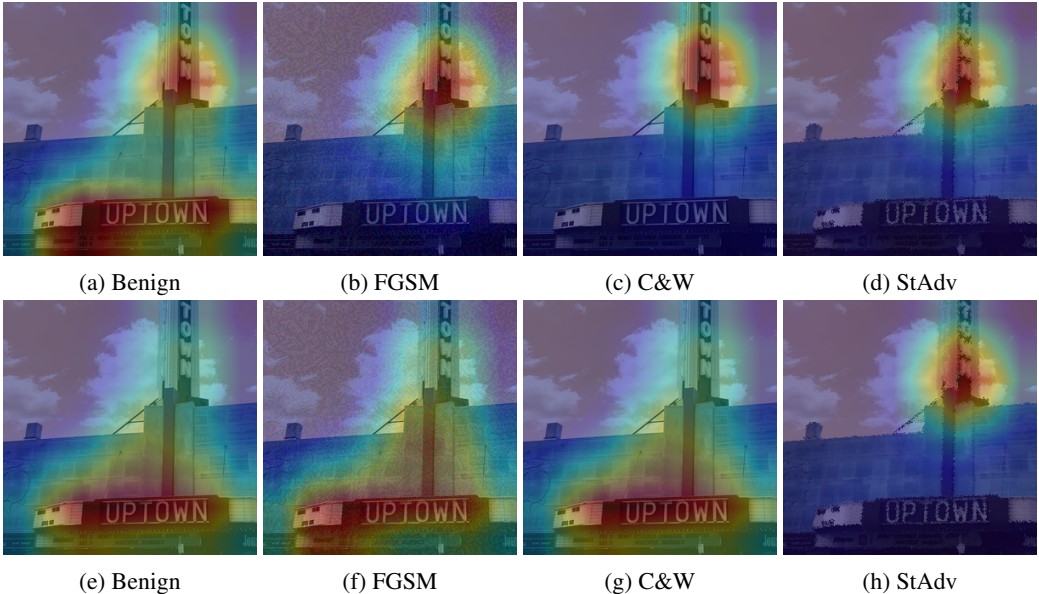

|  |  |  |  |
|---|---|---|---|
| (a) Benign | (b) FGSM | (c) C&W | (d) StAdv |
| (e) Benign | (f) FGSM | (g) C&W | (h) StAdv |

Figure 9: CAM attention visualizations for ImageNet inception_v3 model. The first column shows the CAM maps corresponding to the original images. Column 2-4 show the adversarial examples generated by different methods. The visualizations are drawn for Row 1 (inception_v3 model) and Row 2 (adversarial trained inception_v3 model). (a) and (e)-(g) are labeled as the ground truth "cinema", while (b)-(d) and (h) are labeled as the adversarial target "missile."

We also analyze the attack success rate of these examples under existing defense methods and demonstrate they are harder to defend against, which opens new directions for developing more robust defense algorithms. Finally, we visualize the attention regions of DNNs on our adversarial examples to better understand this new attack.

ACKNOWLEDGMENTS

We thank Zhuang Liu, Richard Shin, Kun Jin, Armin Sarabi and George Philipp for their valuable discussions on this work. This work was supported in part by Berkeley Deep Drive, the Center for Long-Term Cybersecurity, and FORCES (Foundations Of Resilient CybEr-Physical Systems), which receives support from the National Science Foundation (NSF award numbers CNS-1238959, CNS-1238962, CNS-1239054, CNS-1239166), and NSF under grants CNS-1422211 and CNS-1616575.

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

## A    MODEL ARCHITECTURES

Table 4: Architecture of models applied on MNIST

| A | B | C |
|---|---|---|
| Conv(64,5,5) + Relu | Conv(64,8,8) + Relu | Conv(128,3,3) + Relu |
| Conv(64,5,5) + Relu | Dropout(0.2) | Conv(64,3,3) + Relu |
| Dropout(0.25) | Conv(128, 6, 6) + Relu | Dropout(0.25) |
| FC(128) + Relu | Conv(128, 5, 5) + Relu | FC(128) + Relu |
| Dropout(0.5) | Dropout(0.5) | Dropout(0.5) |
| FC(10) + Softmax | FC(10) +Softmax | FC(10)+Softmax |

## B    ANALYSIS FOR MEAN BLUR DEFENSE

Here we evaluated adversarial examples generated by stAdv against the $3 \times 3$ average pooling restoration mechanism suggested in Li & Li (2016). Table 5 shows the classification accuracy of recovered images after performing $3 \times 3$ average pooling on different models.

Table 5: Performance of adversarial examples against the mean blur defense strategy with $3 \times 3$ mean filter.

| Accuracy on recovered images | MNIST | | | CIFAR-10 | |
|---|---|---|---|---|---|
| | A | B | C | Resnet32 | wide ResNet34 |
| $3 \times 3$ Average Filter | 59.00% | 64.22% | 79.71% | 45.12% | 50.12% |

## C    ADVERSARIAL EXAMPLES FOR AN IMAGENET-COMPATIBLE SET, MNIST, AND CIFAR-10

**Experiment settings.** In the following experiments, we perform a grid search of hyper-parameter $\tau$ so that the adversarial examples can attack the target model with minimal deformation. Values of $\tau$ are searched from 0.0005 to 0.05.

**ImageNet-compatible.** We use benign images from the DEV set from the NIPS 2017 targeted adversarial attack competition.[4] This competition provided a dataset compatible with ImageNet and containing target labels for a targeted attack. We generate targeted adversarial examples for the target inception_v3 model. In Figure 10 below, we show the original images on the left with the correct label, and we show adversarial examples generated by stAdv on the right with the target label.

**MNIST.** We generate adversarial examples for the target Model B. In Figure 11, we show original images with ground truth classes 0–9 in the diagonal, and we show adversarial examples generated by stAdv targeting the class of the original image within that column.

**CIFAR-10.** We generate adversarial examples for the target ResNet-32 model. In Figure 12, we show the original images in the diagonal, and we show adversarial examples generated by stAdv targeting the class of the original image within that column.

Table 6 shows the magnitude of the generated flow regarding total variation (TV) and $\mathcal{L}_2$ distance on the ImageNet-compatible set, MNIST, CIFAR-10, respectively. These metrics are calculated by the following equations, where $n$ is the number of pixels:

$$\text{TV} = \sqrt{\frac{1}{n} \sum_{p}^{all\ pixels} \sum_{q \in \mathcal{N}(p)} ||\Delta u^{(p)} - \Delta u^{(q)}||_2^2 + ||\Delta v^{(p)} - \Delta v^{(q)}||_2^2}. \tag{5}$$

---

[4]`https://github.com/tensorflow/cleverhans/tree/master/examples/nips17_adversarial_competition/dataset`

$$\mathcal{L}_2 = \sqrt{\frac{1}{n} \sum_{p}^{all\ pixels} ||\Delta u^{(p)}||_2^2 + ||\Delta v^{(p)}||_2^2} \qquad (6)$$

Table 6: Evaluation Metric (the number in bracket is image size)

| Metric | ImageNet-compatible (299x299) | MNIST (28x28) | CIFAR-10 (32x32) |
|---|---|---|---|
| flow TV | $2.85 \times 10^{-4} \pm 7.28 \times 10^{-5}$ | $8.26 \times 10^{-3} \pm 4.95 \times 10^{-3}$ | $2.21 \times 10^{-3} \pm 1.26 \times 10^{-3}$ |
| flow $\mathcal{L}_2$ | $2.11 \times 10^{-4} \pm 5.19 \times 10^{-5}$ | $5.18 \times 10^{-2} \pm 5.66 \times 10^{-2}$ | $2.76 \times 10^{-3} \pm 2.31 \times 10^{-3}$ |

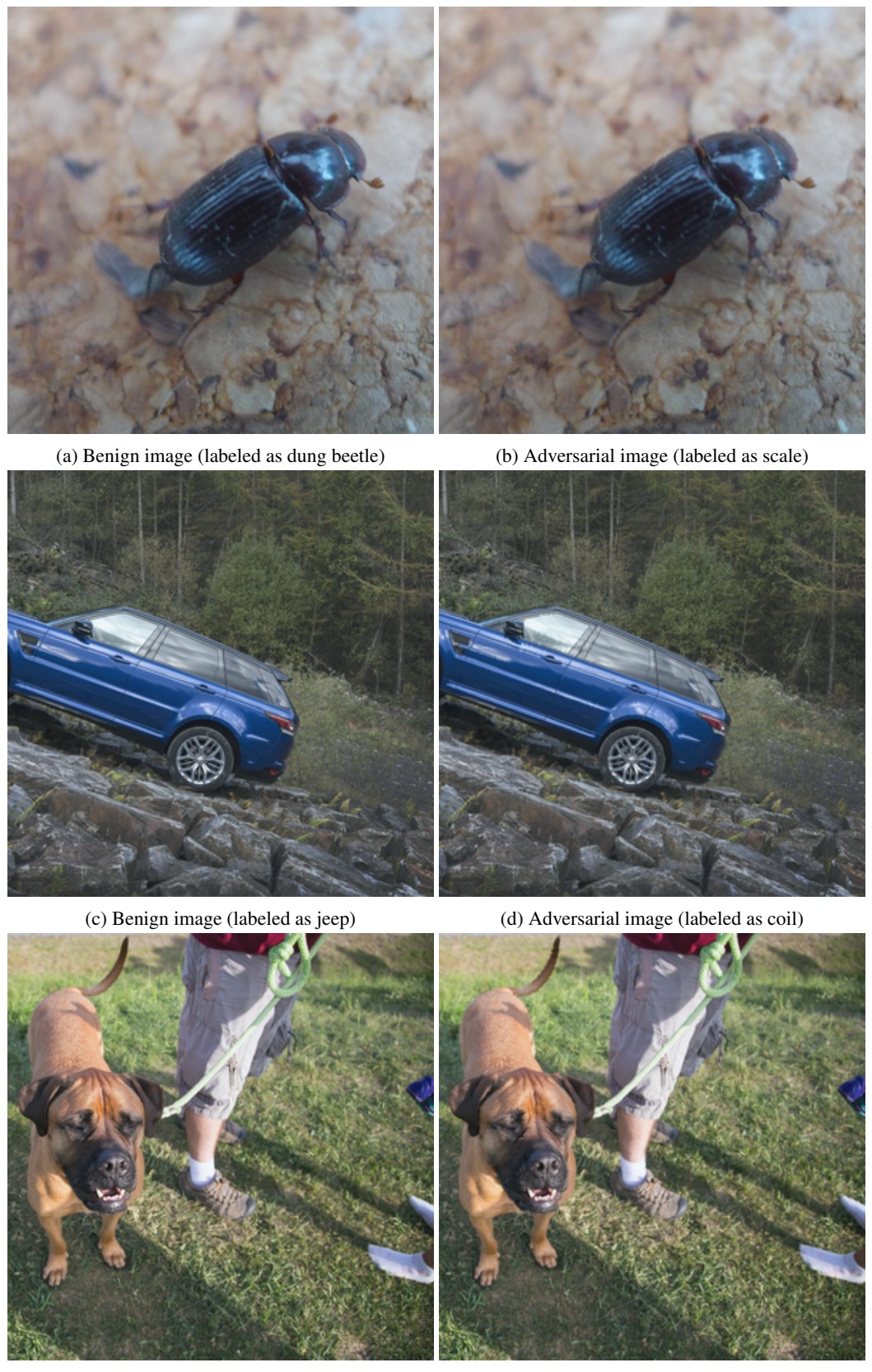

(a) Benign image (labeled as dung beetle)    (b) Adversarial image (labeled as scale)

(c) Benign image (labeled as jeep)    (d) Adversarial image (labeled as coil)

(e) Benign image (labeled as bull mastiff)    (f) Adversarial image (labeled as American lobster)

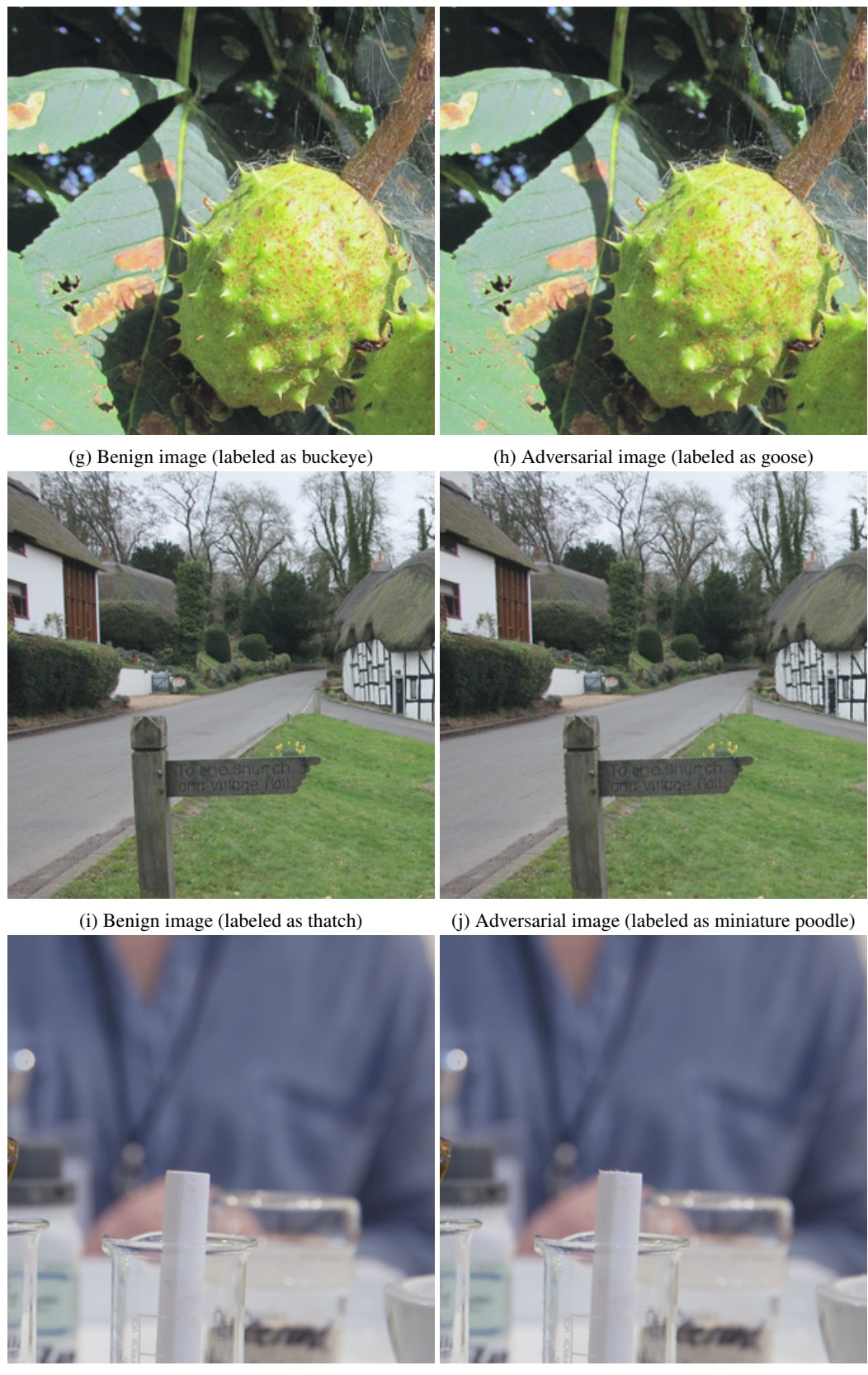

(g) Benign image (labeled as buckeye)      (h) Adversarial image (labeled as goose)

(i) Benign image (labeled as thatch)      (j) Adversarial image (labeled as miniature poodle)

(k) Benign image (labeled as beaker)      (l) Adversarial image (labeled as padlock)

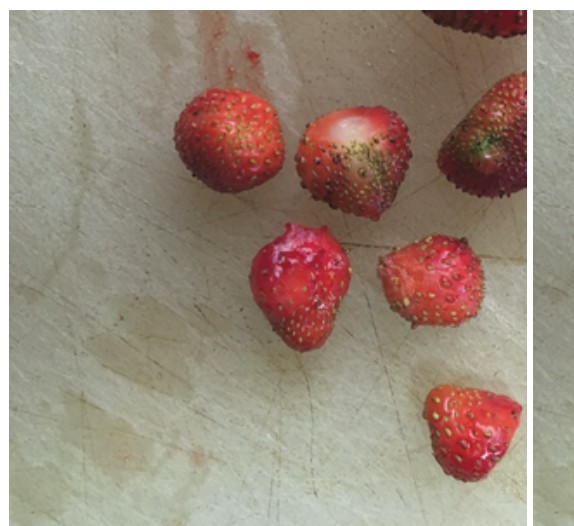

(m) Benign image (labeled as strawberry)

(n) Adversarial image (labeled as tench)

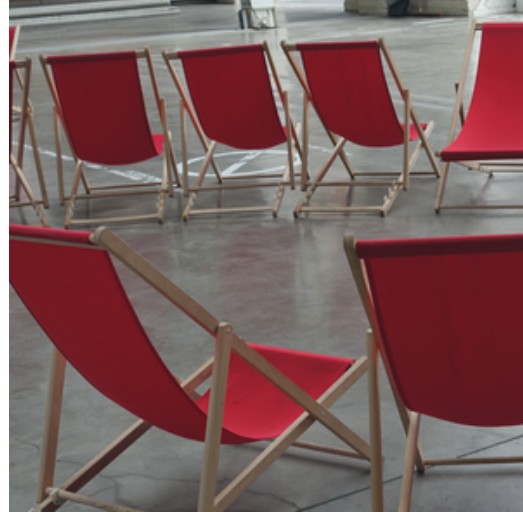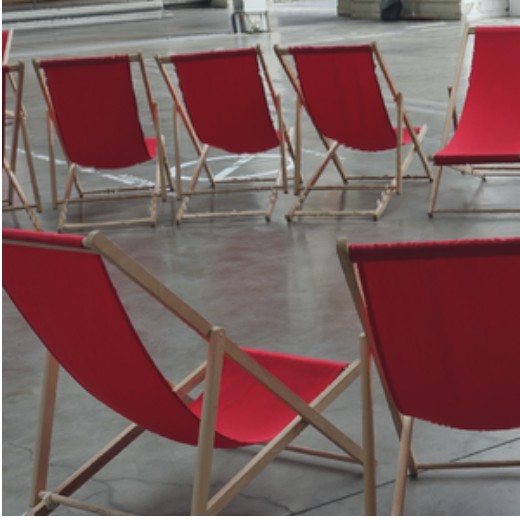

(o) Benign image (labeled as folding chair)

(p) Adversarial image (labeled as power drill)

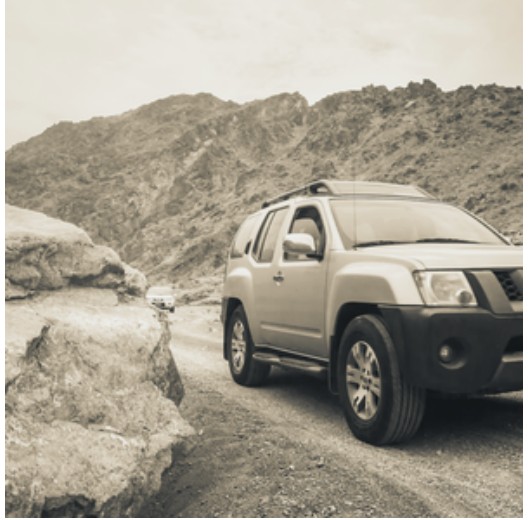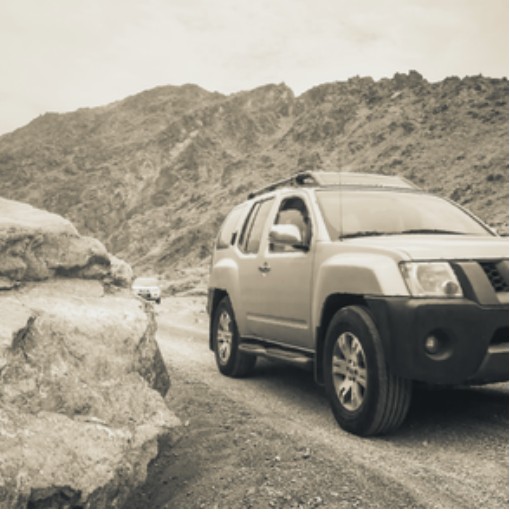

(q) Benign image (labeled as jeep)

(r) Adversarial image (labeled as house finch)

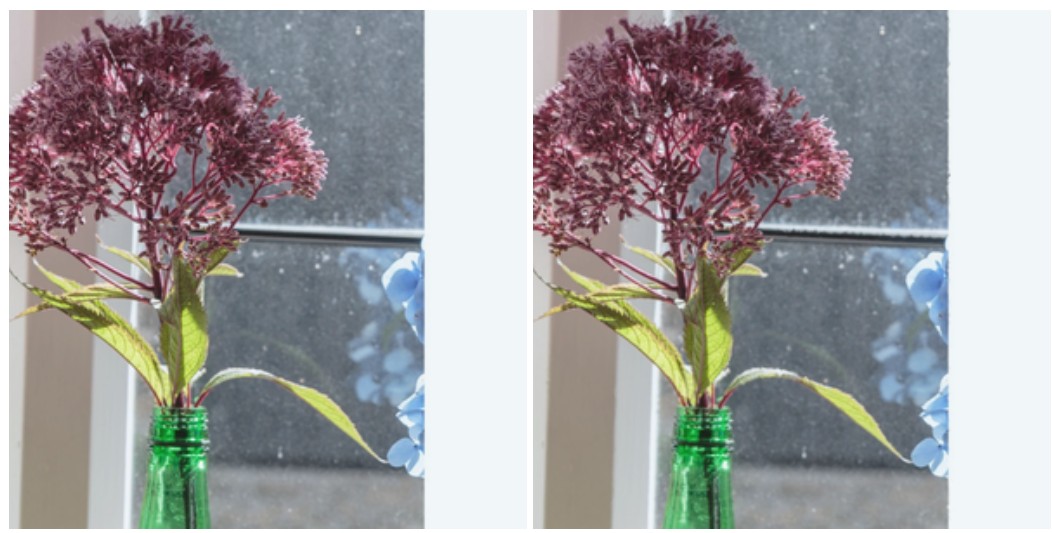

(s) Benign image (labeled as vase)                    (t) Adversarial image (labeled as marmoset)

Figure 10: Examples from an ImageNet-compatible set. Left: original image; right: adversarial image generated by stAdv against inception_v3.

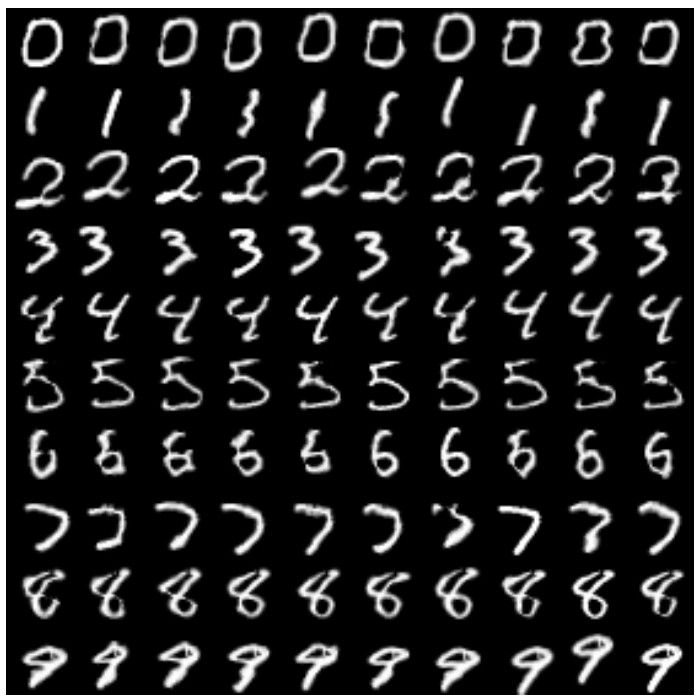

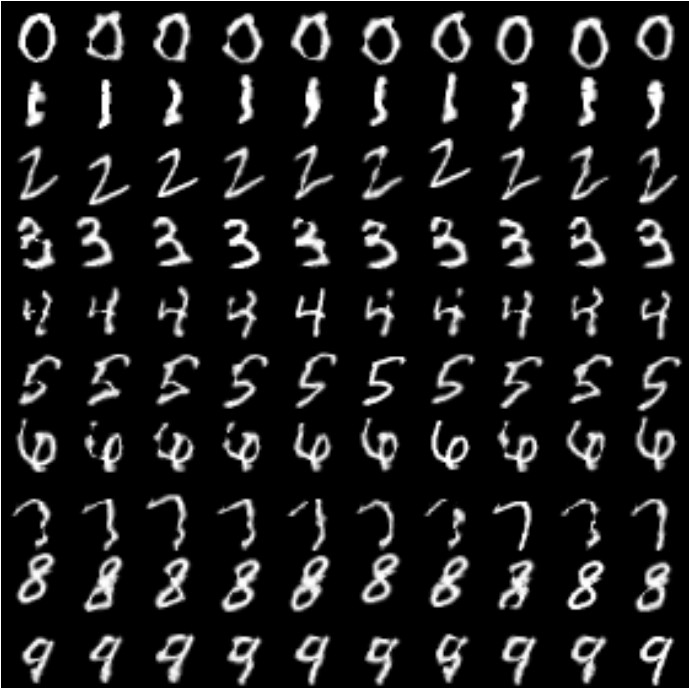

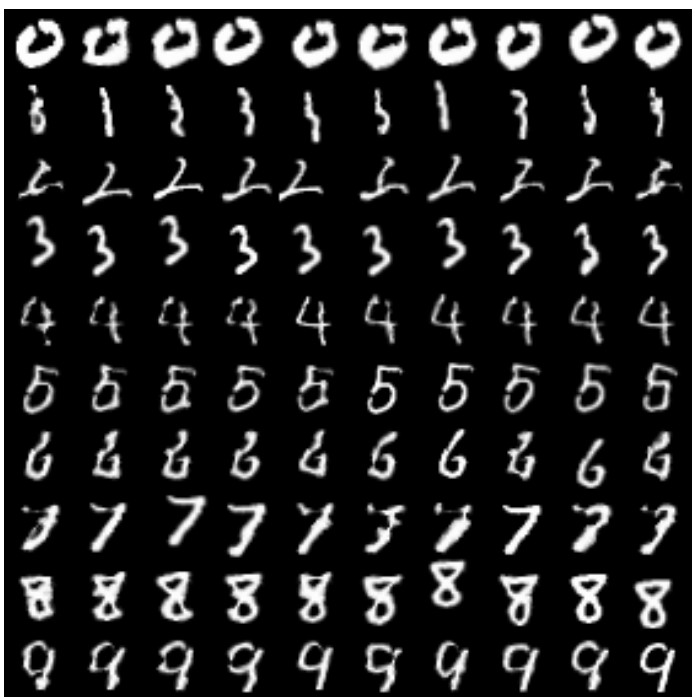

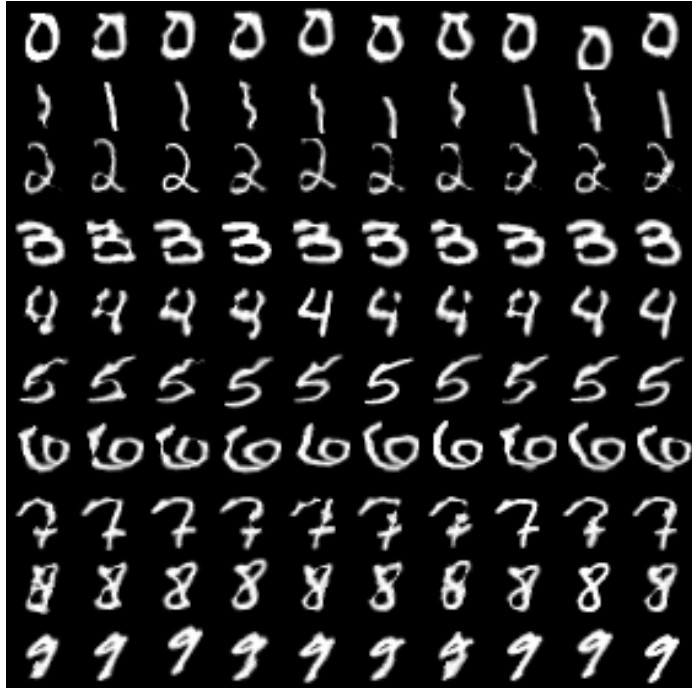

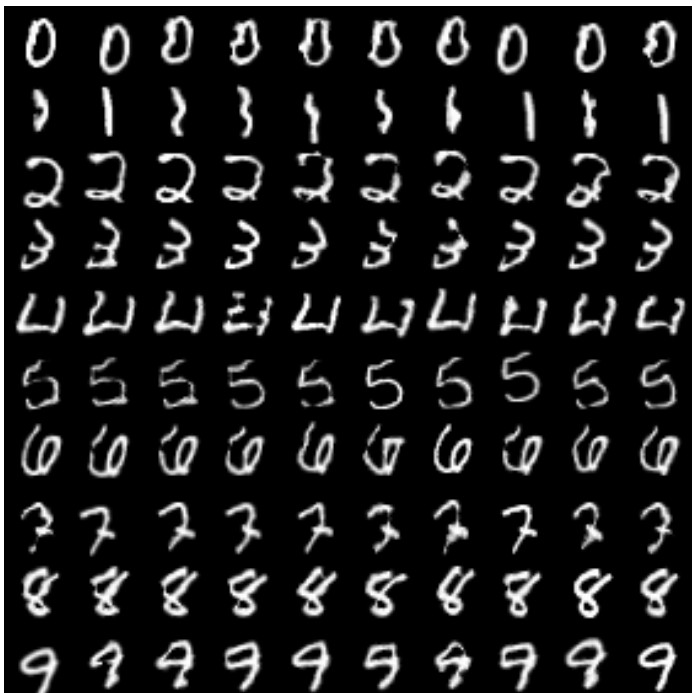

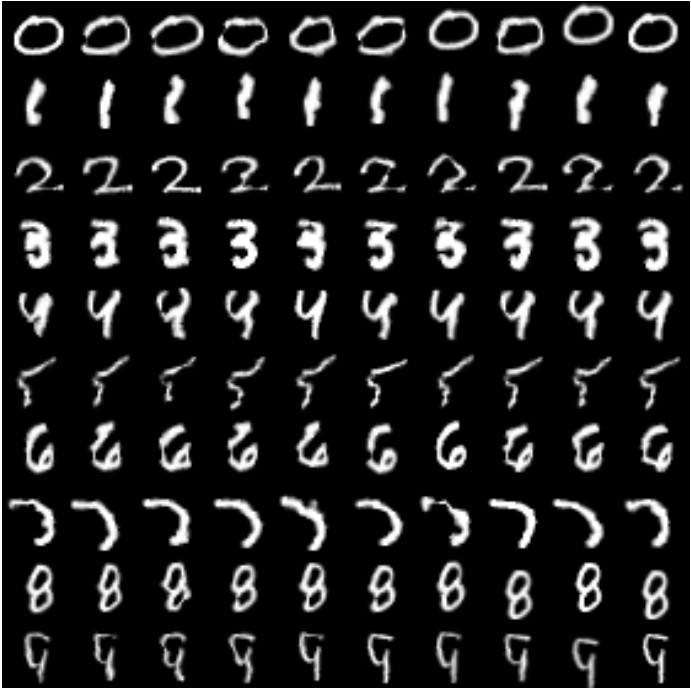

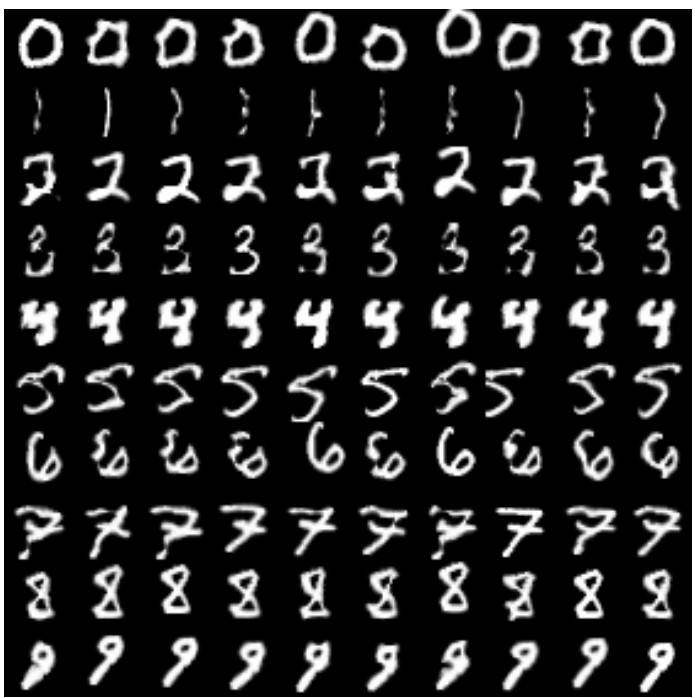

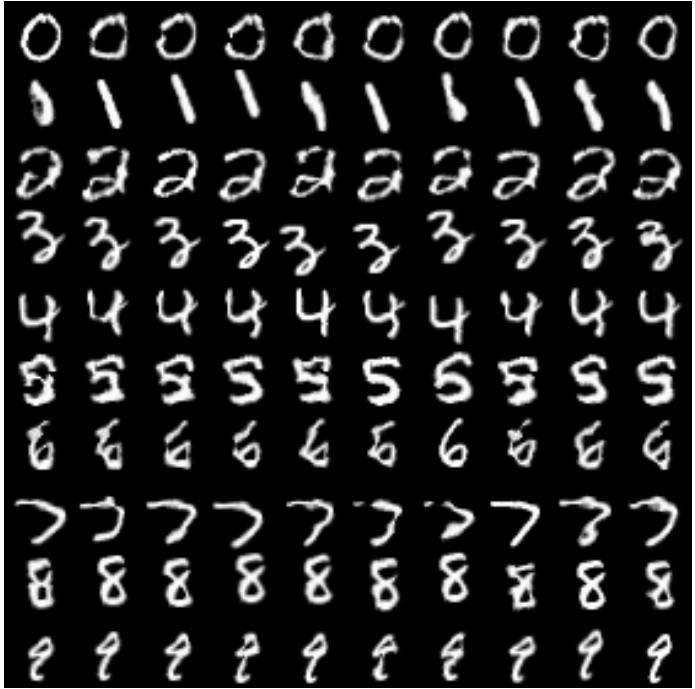

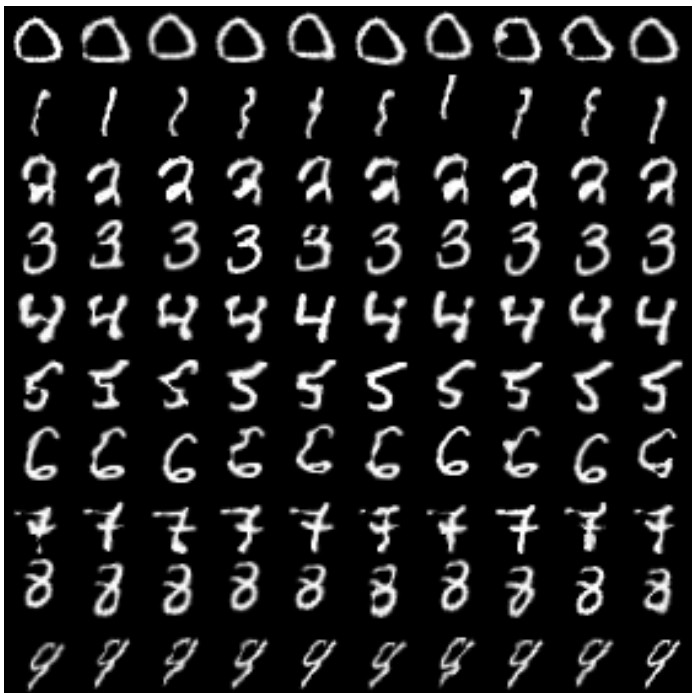

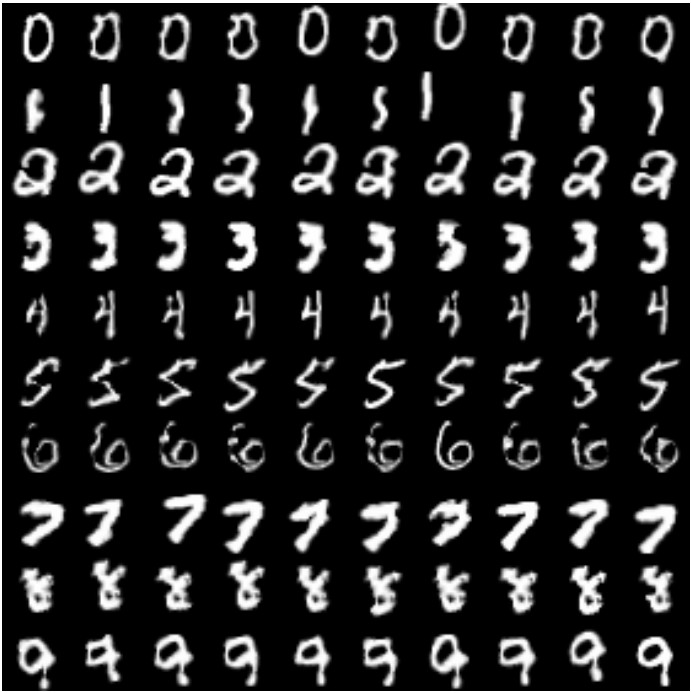

Figure 11: Adversarial examples generated by stAdv against Model B on MNIST. The original images are shown in the diagonal; the rest are adversarial examples that are classified into the same class as the original image within that column.

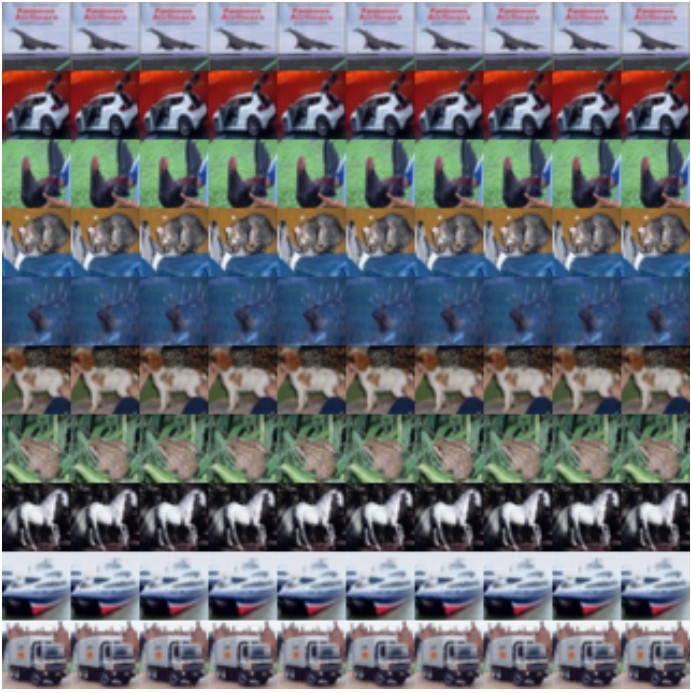

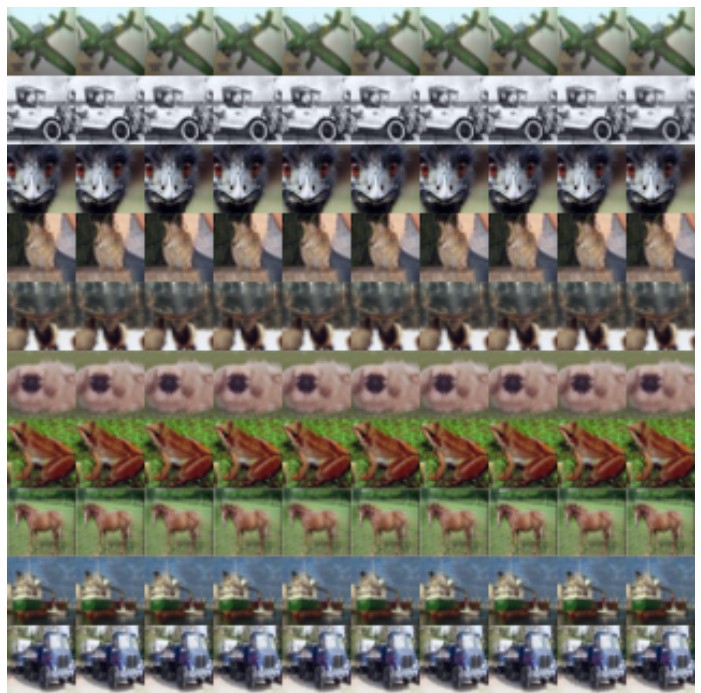

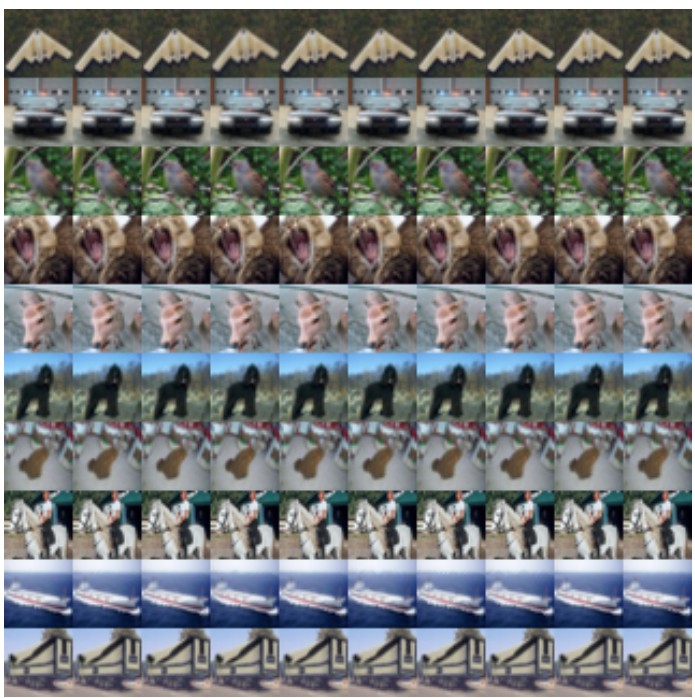

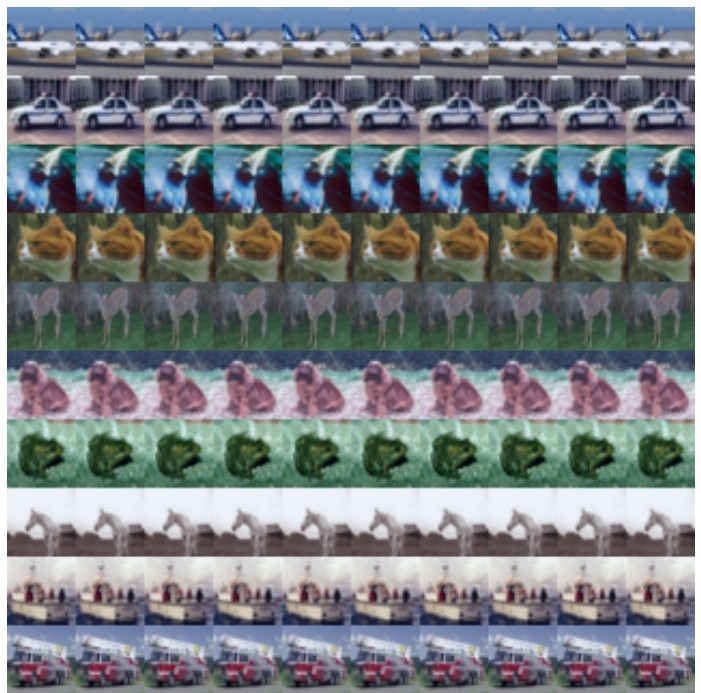

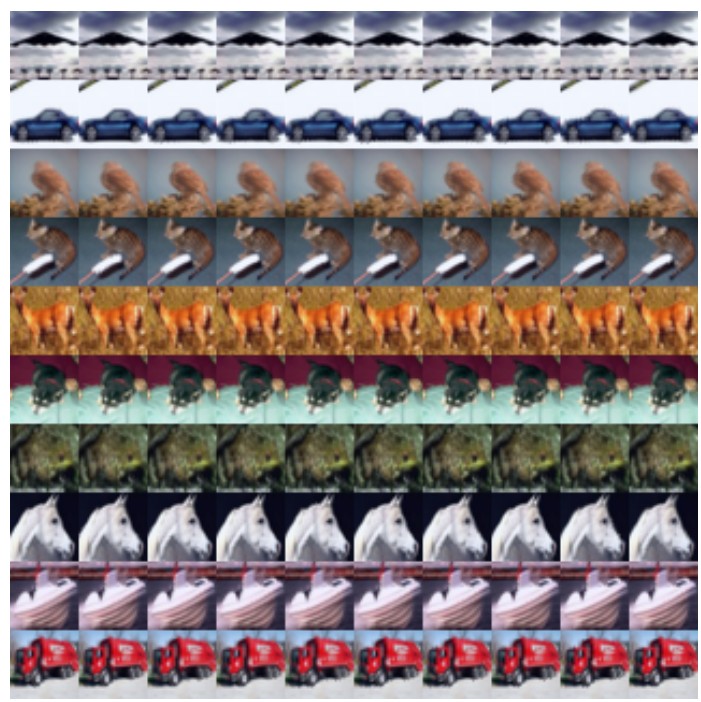

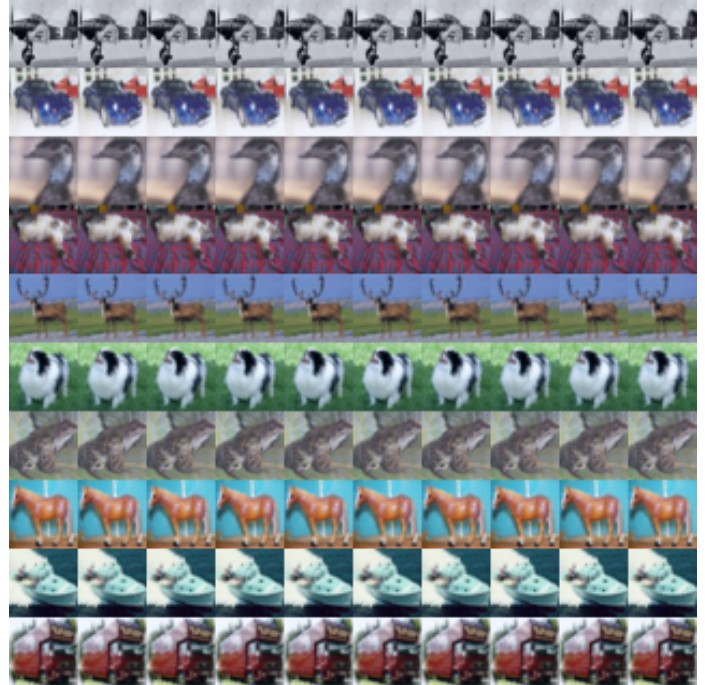

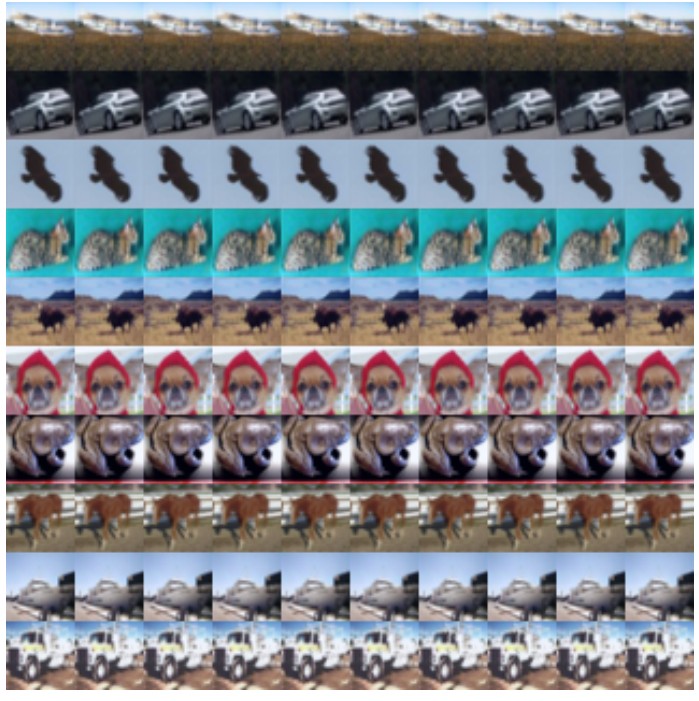

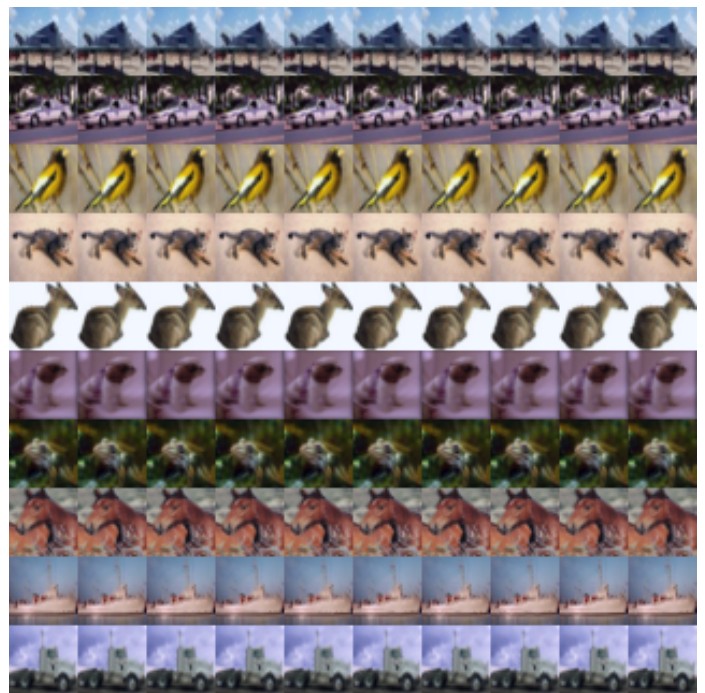

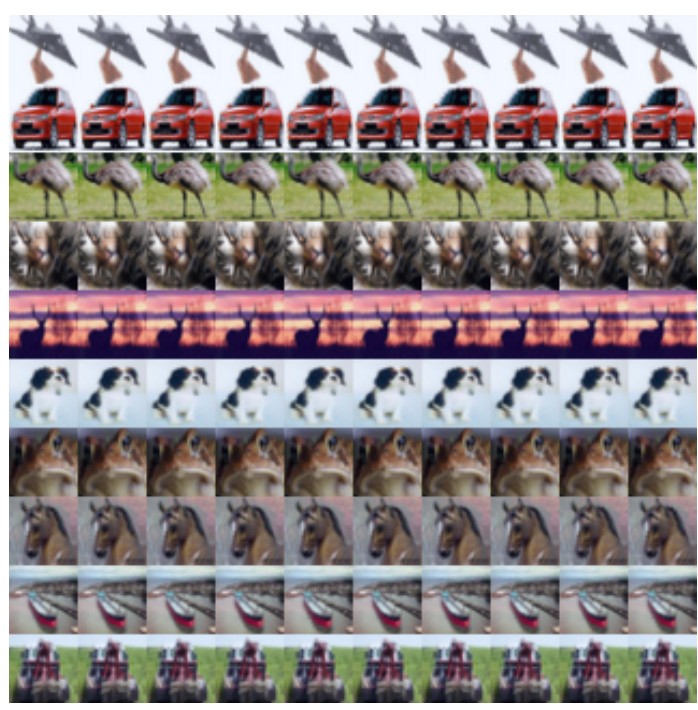

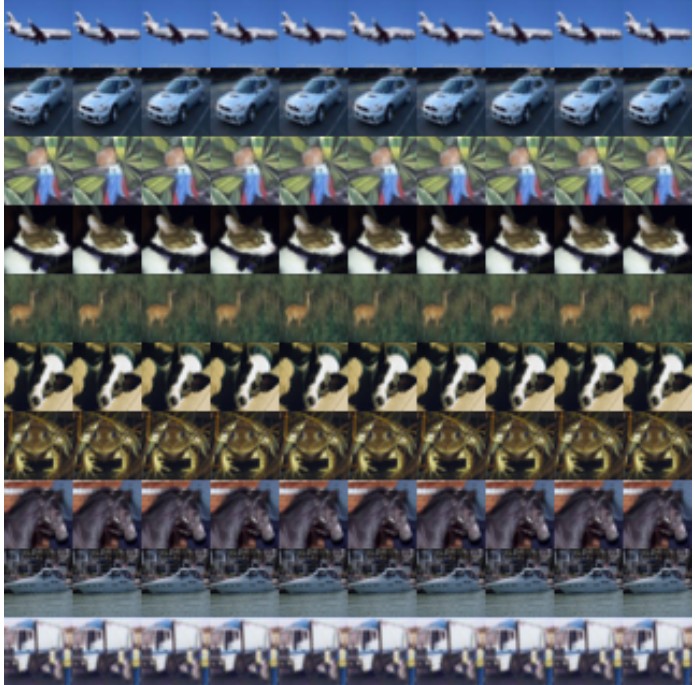

Figure 12: Adversarial examples generated by stAdv against a ResNet-32 on CIFAR-10. The original images are shown in the diagonal; the rest are adversarial examples that are classified into the same class as the original image within that column.

