# OpenReview forum: "Spatially Transformed Adversarial Examples"
_ICLR.cc/2018/Conference — Accept (Poster)_

### Official Review · AnonReviewer3 · 2017-11-26
**An interesting way of creating better adversarial examples.**

**Rating:** 7
**Confidence:** 4

**Review:**

This paper explores a new way of generating adversarial examples by slightly morphing the image to get misclassified by the model. Most other adversarial example generation methods tend to rely on generating high frequency noise patterns based by optimizing the perturbation on an individual pixel level. The new approach relies on gently changing the overall image by computing a flow an spatially transforming the image according to that flow. An important advantage of that approach is that the new attack is harder to protect against than to previous attacks according to the pixel based optimization methods.

The paper describes a novel model method that might become a new important line of attack. And the paper clearly demonstrates the advantages of this attack on three different data sets.

A minor nitpick: the "optimization based attack (Opt)" was first employed in the original "Intriguing Properties..." 2013 paper using box-LBFGS as the method of choice predating FGSM.

---

> ### Author Response · Authors · 2018-01-05
> **reply to "An interesting way of creating better adversarial examples. "**
>
> We thank the reviewer for the constructive suggestions! We have updated the method Opt to C&W throughout the paper.

---

### Official Review · AnonReviewer1 · 2017-11-28
**Interesting solid paper, could use a few more experiments**

**Rating:** 7
**Confidence:** 4

**Review:**

This paper creates adversarial images by imposing a flow field on an image such that the new spatially transformed image fools the classifier. They minimize a total variation loss in addition to the adversarial loss to create perceptually plausible adversarial images, this is claimed to be better than the normal L2 loss functions.

Experiments were done on MNIST, CIFAR-10, and ImageNet, which is very useful to see that the attack works with high dimensional images. However, some numbers on ImageNet would be helpful as the high resolution of it make it potentially different than the low-resolution MNIST and CIFAR.

It is a bit concerning to see some parts of Fig. 2. Some of Fig. 2 (especially (b)) became so dotted that it no longer seems an adversarial that a human eye cannot detect. And model B in the appendix looks pretty much like a normal model. It might needs some experiments, either human studies, or to test it against an adversarial detector, to ensure that the resulting adversarials are still indeed adversarials to the human eye. Another good thing to run would be to try the 3x3 average pooling restoration mechanism in the following paper:

Xin Li, Fuxin Li. Adversarial Examples Detection in Deep Networks with Convolutional Filter Statistics . ICCV 2017.

to see whether this new type of adversarial example can still be restored by a 3x3 average pooling the image (I suspect that this is harder to restore by such a simple method than the previous FGSM or OPT-type, but we need some numbers).

I also don't think FGSM and OPT are this bad in Fig. 4. Are the authors sure that if more regularization are used these 2 methods no longer fool the corresponding classifiers?

I like the experiment showing the attention heat maps for different attacks. This experiment shows that the spatial transforming attack (stAdv) changes the attention of the classifier for each target class, and is robust to adversarially trained Inception v3 unlike other attacks like FGSM and CW.

I would likely upgrade to a 7 if those concerns are addressed.

After rebuttal: I am happy with the additional experiments and would like to upgrade to an accept.

---

> ### Author Response · Authors · 2018-01-05
> **reply to "Interesting solid paper, could use a few more experiments"**
>
> We thank the reviewer for the thoughtful comments and suggestions.
>
> Human study:
> We have added a human study in Section 4.3.  In particular, we follow the same perceptual study protocol used in prior image synthesis work [Zhang et al. 2016; Isola et al. 2017]. In our study, the participants are asked to choose the more visually realistic image between (1) an adversarial example generated by stAdv and (2) its original image. The user study shows that the generated adversarial examples can fool human participants 47% of the time (perfectly realistic results would achieve 50%). This experiment shows that our adversarial examples are almost indistinguishable from natural images. Please see section 4.3 for more details.
>
> 3x3 mean blur defense
> We included the suggested related work and added an analysis of the 3x3 average pooling restoration mechanism [Li et al. 16’]. See section 4.4 and Table 5 in Appendix B for the discussion and results. In summary, the restoration is not as effective on stAdv examples. The classification accuracy on restored stAdv examples is around 50% (Table 5), compared to restored C&W examples (around 80%) and FGSM examples (around 70%)  [Carlini et al. 2017, Li et al. 2016]. In addition, stAdv achieves near 100% attack success rate in a perfect knowledge adaptive attack [Carlini et al. 2017].
>
> Comparison with C&W and FGSM (Figure 4)
> In our revised version, we have updated Figure 4 to show adversarial examples for FGSM and C&W with a strong adversarial budget as: L_infinity perturbation limit of 0.3 on MNIST and 8 on CIFAR-10. We apply the same setting for the later evaluations against defenses
>
> References:
> [Carlini et al. 2017] Carlini, Nicholas, and David Wagner. "Adversarial Examples Are Not Easily Detected: Bypassing Ten Detection Methods." arXiv preprint arXiv:1705.07263 (2017).
> [Li et al. 2016] Li, Xin, and Fuxin Li. "Adversarial examples detection in deep networks with convolutional filter statistics." arXiv preprint arXiv:1612.07767 (2016).
> [Zhang et al. 2016] Zhang, Richard, Phillip Isola, and Alexei A. Efros. "Colorful image colorization." European Conference on Computer Vision. Springer International Publishing, 2016.
> [Isola et al. 2017]Isola, Phillip, et al. "Image-to-image translation with conditional adversarial networks." arXiv preprint arXiv:1611.07004 (2016).

---

### Official Review · AnonReviewer2 · 2017-12-01
**Moving the pixels to get adversarial attacks is also possible**

**Rating:** 9
**Confidence:** 5

**Review:**

This paper proposes a new way to create adversarial examples. Instead of changing pixel values they perform spatial transformations.

The authors obtain a flow field that is optimized to fool a target classifier. A regularization term controlled by a parameter tau is ensuring very small visual difference between the adversarial and the original image.

The used spatial transformations are differentiable with respect to the flow field (as was already known from previous work on spatial transformations) it is easy to perform gradient descent to optimize the flow that fools classifiers for targeted and untargeted attacks.

The obtained adversarial examples seem almost imperceivable (at least for ImageNet).
This is a new direction of attacks that opens a whole new dimension of things to consider.

It is hard to evaluate this paper since it opens a new direction but the authors do a good job using numerous datasets, CAM attention visualization and also additional materials with high-res attacks.

This is a very creative new and important idea in the space of adversarial attacks.

Edit: After reading the other reviews , the replies to the reviews and the revision of the paper with the human study on perception, I increase my score to 9. This is definitely in the top 15% of ICLR accepted papers, in my opinion.

Also a remark: As far as I understand, a lot of people writing comments here have a misconception about what this paper is trying to do: This is not about improving attack rates, or comparing with other attacks for different epsilons, etc.
This is a new *dimension* of attacks. It shows that limiting l_inf of l_2 is not sufficient and we have to think of human perception to get the right attack model. Therefore, it is opening a new direction of research and hence it is important scholarship. It is asking a new question, which is frequently more important than improving performance on previous benchmarks.

---

> ### Author Response · Authors · 2018-01-05
> **reply to "Moving the pixels to get adversarial attacks is also possible"**
>
> We thank the reviewer for the helpful comments. To further improve our work, we have added a user study to our updated version to evaluate the perceptual realism for the generated instances. In particular, we follow the same perceptual study protocol used in prior image synthesis work [Zhang et al. 2016; Isola et al. 2017]. In our study, the participants are asked to choose the more visually realistic image between (1) an adversarial example generated by stAdv and (2) its original image. The user study shows that the generated adversarial examples can fool human participants 47% of the time (perfectly realistic results would achieve 50%). This experiment shows that our adversarial examples are almost indistinguishable from natural images. Please see section 4.3 for more details.

---

### Public Comment · (anonymous) · 2017-10-30
**Visibility-fooling effectiveness tradeoff not so clear**

Thanks for the interesting paper -- it opens a new axis for adversarial attacks.
While the idea is interesting, the tradeoff between visibility and fooling effectiveness is not entirely clear.
The potential greatest concern that I have about this paper is that the perturbations are way too visible.

For claiming an adversarial perturbation to be superior to the others, not only fooling rates (or ``attack success rates'') but also perceptual similarity to the original image should be compared. Since there is no standard measure for the latter, people have resorted to Lp distances for additive perturbations. One cannot do the same in this paper because the perturbations are not additive, as the authors have described: "for stAdv, we cannot use Lp norm to bound the distance as translating a image by one pixel may introduce large Lp penalty."

The paper still compares the attack success rate comparisons against previous attacks in table 2. The authors have used bare-eye observation to set the operating point: "We instead constrain the spatial transformation flow and show that our adversarial examples have high perceptual quality in Figures 2, 3, and 4." But this is hardly convincing, since the perturbations seem to be _always visible_ if one looks closely (fractal patterns along edges), often more visible than previous additive attacks. The claim "geometric changes are small and locally smooth" is hard to buy.

Also the number of visual examples is too stingy for ImageNet (only 2). Given their importance, there should be at least 10 examples per dataset. Also, visualisations in this paper are too low resolution in general and so many image artifacts may be overlooked.

I would suggest the following for making a more convincing case for the paper:
Visually compare the _minimal_ perturbation needed for a successful attack, on MNIST, CIFAR, and ImageNet, with at least 10 examples in _high resolution_ figures.

---

> ### Author Response · Authors · 2017-11-03
> **Reply to "Visibility-fooling effectiveness tradeoff not so clear"**
>
> We thank the commenter for the suggestions. We are glad that the commenter finds the idea interesting and thinks it opens a potential direction. We agree that it is hard to evaluate the perceptual similarity of adversarial examples and that L_p may not be the best and is not suitable for our proposed method.
>
> We emphasize that the perturbations are almost invisible for CIFAR and ImageNet datasets (we cannot tell the difference in the examples in our paper).We believe these results are more important than the results on MNIST, where the differences are visible because these wiggly/sketchy distortions occur more naturally in handwritten digits.
>
> Based on the suggestion, we have gotten the permission from the Chair to add an anonymized link (https://www.dropbox.com/sh/pl7sbecks6ja5g0/AACVdlRg96heBkICOWl1IQm4a?dl=0) here for ten more image examples on MNIST, CIFAR, and ImageNet in high-resolution figures.
> We will also add more examples in the appendix of our updated version.

---

> > ### Public Comment · ~Xun_Huang1 · 2017-11-08
> > **Tradeoff by changing τ?**
> >
> > Is it possible to achieve visibility-fooling tradeoff by changing τ in equation (2)? The weight of  L_{flow} should control the visibility of the attack.

---

> > > ### Author Response · Authors · 2017-11-10
> > > **Reply to "Tradeoff by changing τ“**
> > >
> > > you are right, the \tau here is used to control the tradeoff between visibility-fooling.
> > > We apply grid search to tune \tau, and the corresponding results are shown in the appendix((https://www.dropbox.com/sh/pl7sbecks6ja5g0/AACVdlRg96heBkICOWl1IQm4a?dl=0).

---

### Public Comment · (anonymous) · 2017-11-12
**state-of-the-art attacks and defenses are not compared.**

I agree with the previous comments that this paper proposes a new method to generate adversarial examples. However, why this new method is needed is not well supported.

1. Is your goal to have a higher attack success rate?

    1a. The evaluations did not really show the new method has higher attack success rates than the state-of-the-art attacks. In particular, what is exactly Opt attack? the attack proposed by Carlini and Wagner has very high attack success rate (close to 100%) for adverarially trained neural networks. You can refer to these two papers:

https://openreview.net/pdf?id=BkpiPMbA-
https://arxiv.org/abs/1709.05583

    1b. The paper did not evaluate their attacks against state-of-the-art defense methods. Adversarial training is not state-of-the-art defense. State-of-the-art defense leverages neighborhood around an instance to predict its label, instead of the single instance alone. It would be interesting to show whether the new attacks are effective against state-of-the-art defense, e.g., https://arxiv.org/abs/1709.05583


 2. is your goal to generate more visually realistic adversarial examples? If this is the goal of the paper, I think the authors should provide user studies to evaluate whether their adversarial examples are more visually realistic than existing adversarial examples. Showing several examples is not sufficient.

---

> ### Public Comment · (anonymous) · 2017-11-12
> **region based method https://arxiv.org/abs/1709.05583**
>
> I cannot see that the paper: https://arxiv.org/abs/1709.05583 is a state-of-art defense (or better than adversarial training). Only CW attack is considered in that paper (e.g., no FGSM), experiments are not enough to support this argument.
>
> Meanwhile, this method is more like adding random noises around the each clean image to obtain multiple images for classification (ensemble classification). I cannot see this method can be robust to FGSM with a high value of sigma.

---

> ### Author Response · Authors · 2017-11-23
> **Reply to "state-of-the-art attacks and defenses are not compared"**
>
> The main reason that stAdv is needed or our main goal of the paper is to provide a new way to think about adversarial examples: where an attacker can move pixels by some amount, instead of adding or subtracting pixel values. Investigating this unexplored side of adversarial examples is worthwhile, even if it would involve attacks that are strictly weaker, although that is not the case here. In this paper, we focused on showing the differences between spatially transformed adversarial examples and additive adversarial examples in terms of attack robustness analysis and attention visualization.
>
> High attack success rate and visually realistic adversarial examples are goals of lower priority.
> [1a] We thank the comment for their enthusiasm in this new attack and for bringing up the desire for additional success rate comparisons with additive adversarial examples and on more defenses. One result we didn’t include in the paper since it is not the main focus as we mention above is that we can also achieve 100% attack success rate on white-box attacks on adversarially trained models, like other optimization attacks. We can of course add these results in our updated version.
>
> The Opt attack is C&W’s attack (last sentence in the first paragraph of Intro) and we will clarify this in our updated version.
>
> [1b]  We are definitely interested in further testing our attack method against other state-of-the-art defenses and thank you for the suggestions. One of the two papers mentioned is a concurrent ICLR 2018 submission. The other one is an arxiv paper published in mid-September. It is impossible to directly apply these defenses in our submission. Moreover,  the second reference has shown that Cao’s defense can already be attacked with ensemble optimization method. We are watching for better defense methods, but to the best of our knowledge currently, the most efficient methods are the adversarial training based methods we tested.
>
> [2] As the commenter mentions about the visual quality, yes, we did intend to show that the adversarial examples generated this way are more realistic. We will conduct a user experiment, which the commenter also suggested.  In addition, supporting the claim that the examples are realistic will create useful evidence that that the L_p norm is not a good measurement. The vision community has long since noticed this weakness of the L_p norm, but no better ones are provided. Here we actually show that with relatively high L_p norm, we can still generate perceptually realistic adversarial examples, which raises up an open research direction to propose better distance measurements between adversarial examples and normal instances.

---

### Public Comment · (anonymous) · 2017-12-03
**Interesting direction for adversarial attack, but the optimization based attack results are not convincing**

The paper provides new type of adversarial attack which is different from the previous works.
However, the optimization based attack results (CW attacks) shown in table 3 are very suspicious considering tha fact that the attacks are performed in white box attack assumption.
There are a lot of papers showing iterative/optimization based attacks are much more stronger than FGSM method for the networks trained with standard training or adversarial training.
CW L_infinity attacks are known to be sensitive to the hyper parameters (tau, learning rate, constant c).
And those hyper parameters have to be optimally chosen per "each" example.
The authors should provide convincing explanation why they got such poor results for CW attack (even lower attack rates compared to FGSM for model C ).
Unless they provide reasonable explanation for that, the paper remains suspicious and claims wrong conclusion based on unfair comparison.

---

> ### Author Response · Authors · 2018-01-10
> **reply to "Interesting direction for adversarial attack..."**
>
> We thank the commenter for the question.
> First, in Table 3, the setting is: we generate adversarial examples against the undefended model based on different methods, and then we test them against adversarially trained models to see how well the examples transfer to defended models. However, the goal of this experiment is not to compare which attack is more robust; it was to test the hypothesis that retraining with additive adversarial examples would have a different effect on stAdv examples, compared to other additive adversarial examples. As for our results on the additive adversarial examples, they are consistent with previous results showing that, in similar experiments, C&W examples [Fig 10 in Carlini & Wagner] are less transferable than FGSM examples [Fig 3(b) in Papernot & McDaniel].
> Second, we didn’t tune parameters for C&W. Instead we use the default setting for C&W [1] and also use the same setting for stAdv. So we think this is fair comparison.
>
> [Carlini, Nicholas, and David Wagner]. "Towards evaluating the robustness of neural networks." Security and Privacy (SP), 2017 IEEE Symposium on. IEEE, 2017.
> [Papernot, Nicolas, and Patrick McDaniel]. "Extending Defensive Distillation." arXiv preprint arXiv:1705.05264(2017).
> [1] https://github.com/carlini/nn_robust_attacks/blob/master/li_attack.py

---

### Public Comment · ~Seyed-Mohsen_Moosavi-Dezfooli1 · 2017-12-13
**Prior works**

I found this work very interesting and the paper is neatly written.

It would be good if the authors comment on the differences between their approach and prior work that previously constructed adversarial examples with spatial transformations (a.k.a. geometric transformations). In particular, these types of adversarial examples have first been considered in a 2015 paper (https://arxiv.org/abs/1507.06535) published in BMVC. Also, there is a master thesis which addresses a very similar setting to this paper (i.e., using a flow field): https://infoscience.epfl.ch/record/230235 (e.g. pages 32-34).

The paper would be stronger if such comparison is provided, IMHO.

---

> ### Author Response · Authors · 2018-01-09
> **reply to "Prior works"**
>
> Thanks for pointing out these two nice works. We were not aware of BMVC 15’ and the master thesis (published on 2017-08-20, updated on 2017-12-13) during our submission. But we have cited them in our revision and we are happy to discuss the differences based on our understanding.
>
> Here are our comments regarding the differences between ours and these works:
>
> BMVC 15’ “Manitest: Are classifiers really invariant?”: This work studies “whether classifiers are invariant to basic transformations such as rotations and translations”. The paper proposes a novel algorithm to compute invariance score for any classifier. Our paper differs from this paper in two ways: First, our goal is to produce adversarial examples that can fool the classifier, visually realistic, and with minimal changes while the global transformation introduces big changes, and sometimes produce unrealistic images. Second, we study the local deformation rather than global transformation as global transformation will often change the image dramatically.
>
> The master thesis “Measuring Robustness of Classifiers to Geometric Transformations” (created on 2017-08-20, modified on 2017-12-13). The latest version (Dec 13, 2017) was updated after our submission. Similar to BMVC 15’, this *concurrent* work studies the transformation invariance for CNNs with a focus on high dimensional transformation. The thesis proposes new methods for measuring the invariance score and studies the invariance score regarding varying depths of the network. Again, we target for a different goal as we aim to produce adversarial examples with locally smooth spatial transformation rather than computing invariance scores. In addition to the difference in the goal, the loss function, the parameterization of the transformation, and the optimization method are all different.

---

### Public Comment · (anonymous) · 2017-12-20
**Transferability and Efficiency of Spatial Transformation Adversarial Method**

This work explores a new direction on generating adversarial examples. However, I would like to share my concerns about the transferability and time efficiency of the spatial transformation method. As you referred in the paper "we will focus on the white box setting...", whether this method can be applied into black box attacks. And it seems like too expensive to solve this optimal problem by L-BFGS. Could you reveal further details about the time consumption of generating adversarial examples and compared results with other methods?

---

> ### Author Response · Authors · 2018-01-09
> **reply to "Transferability and Efficiency of Spatial Transformation Adversarial Method "**
>
> We thank the commenter for the questions.
> Running time:
> To compare stAdv and C&W’ speed, we conduct the following three experiments on CIFAR-10 dataset. The target model is ResNet32.
>  (1). stAdv(LBFGS): \tau 0.05, max_step 200, learning rate 5e-3, solver: lbfgs with line_search.
>  (2)  stAdv(ADAM): \tau 0.05, max_step 200, learning rate 5e-3, solver: ADAM.
>  (3)  C&W: bound linf 8,  initial cost 10e-5, max_step 1000 largest_const 2e+1, confidence 0, const_factor 2.0, solver: ADAM  (We use  the default settings  in C&W’s official GitHub repo.)
> Here we report the average running time over 50 random images for each method:
> StAdv (LBFGS): time: 5.435s, attack success rate : 100%
> StAdv (Adam):  time:  0.092s,  attack success rate : 100%
> C&W (Adam):  time:  4.0214s,  attack success rate : 100%
> The results show that stAdv with LBFGS solver is slightly slower than C&W. However, stAdv with Adam solver is much faster than C&W, and both of them achieve the same success rate. We conclude that speed is not a big issue for stAdv. Also, we note that the solver (Adam vs. LBFGS) plays a more critical role in running time compared to what to optimize (e.g. flow vs. pixels.)
> Blackbox attack:
> Thanks for your suggestions. It is definitely worth exploring. In this work, we focus on the white-box setting to explore what a powerful adversary can do based on the Kerckhoffs’s principle[Shannon, 1949] to better motivate defense methods. Besides, stAdv can achieve transferability based black-box attack. We will leave other blackbox attack techniques as our future work.
>
> Reference:
> [Shannon, 1949] Shannon, Claude E. "Communication theory of secrecy systems." Bell Labs Technical Journal 28.4 (1949): 656-715

---

### Author Response · Authors · 2018-01-08
**Summary of changes to the manuscript**

Changes made in our revised version are listed as below:
- Added human perceptual study of our algorithm in section 4.3.
- Analyzed the efficiency of mean blur defense strategy against our algorithm in section 4.4.
- Added a detailed description of experiment setting for C&W and FGSM method in section 4.4.
- Added more adversarial examples for ImageNet-Compatible set, MNIST, and CIFAR-10 in Appendix C.
- Updated figure 4 with a strong adversarial budget on MNIST and CIFAR-10.
- Fixed some grammatical errors.
- Changed the name “Opt” to “C&W”.

We would like to thank the reviewers again for the useful feedbacks and suggestions.

---

### Public Comment · (anonymous) · 2018-01-15
**Results in Table 3**

Hi,

Thanks for the paper, I really enjoy reading the paper, and I think the idea is novel and really interesting approach on how to create adversarial examples that are perceptually indistinguishable!

I have a question: was the adversarial training experiment conducted in table 3 also includes adversarial images generated by your method as part of the training?

Thanks!

---

> ### Author Response · Authors · 2018-01-19
> **reply to 'results in table 3'**
>
> Thanks for the question! In our adversarial training experiment, we only use adversarial examples generated by FGSM to adversarially retrain. The goal of our experiments here is to see if the proposed spatially transformed adv lies on different data manifold with the existing adversarial examples, instead of showing how robust such attack is.

---

> > ### Public Comment · (anonymous) · 2018-01-20
> > **reply to 'results in table 3'**
> >
> > Thanks for the reply!
> >
> > I see. I think it is a valid argument to say that it lies on different data manifold compared to FGSM and/or CW. Perhaps it can be considered to mention that in the future because I initially misunderstood that you want to show how robust the attack is, and I thought it was not a fair test to compare the adversarial training and ensemble adversarial training if it was tested only against FGSM & CW (especially FGSM, perhaps RAND+FGSM from https://arxiv.org/abs/1705.07204 is more appropriate for comparison, since it has been shown that FGSM is weak against adversarially trained model due to gradient masking).
> >
> > But overall I think the method is really interesting and refreshing!
> >
> > Good luck!

---

### Public Comment · ~Beranger_Dumont1 · 2018-05-11
**new public TensorFlow implementation**

Hi all,

This is to let anyone interested know that we have done a public (and unit-tested) TensorFlow implementation of Spatially Transformed Adversarial Examples. It can be found at https://github.com/rakutentech/stAdv

All comments are welcome!

---

### Decision · Program_Chairs · 2018-01-29
**ICLR 2018 Conference Acceptance Decision**

**Decision:**

Accept (Poster)

**Comment:**

All reviewers gave "accept" ratings.
it seems that everyone thinks this is interesting work.

The paper generated a large number of anonymous comments and these were addressed by the authors.